# The HN protein of Newcastle disease virus induces cell apoptosis through the induction of lysosomal membrane permeabilization

Yu Chen[1,2,3], Shanshan Zhu[1], Tianxing Liao[1], Chunxuan Wang[1], Jiajun Han[1], Zhenyu Yang[1], Xiaolong Lu[1,2,3], Zenglei Hu[2,4], Jiao Hu[1,2,3], Xiaoquan Wang[1,2,3], Min Gu[1,2,3], Ruyi Gao[1,2,3], Kaituo Liu[2,4], Xiaowen Liu[1,2,3], Chan Ding ⓘ[2,5]*, Shunlin Hu[1,2,3]*, Xiufan Liu ⓘ[1,2,3]*

1 Animal Infectious Disease Laboratory, College of Veterinary Medicine, Yangzhou University; Yangzhou, Jiangsu, China, 2 Jiangsu Co-innovation Center for Prevention and Control of Important Animal Infectious Diseases and Zoonosis, Yangzhou University, Yangzhou, China, 3 Jiangsu Key Laboratory of Zoonosis, Yangzhou University, Yangzhou, China, 4 Joint International Research Laboratory of Agriculture and Agri-Product Safety, The Ministry of Education of China, Yangzhou University, Yangzhou, China, 5 Department of Avian Infectious Diseases, Shanghai Veterinary Research Institute, Chinese Academy of Agricultural Sciences, Shanghai, China

* shoveldeen@shvri.ac.cn (CD); slhu@yzu.edn.cn (SH); xfliu@yze.edu.cn (XL)

**Data Availability Statement:** All relevant data are within the manuscript and its Supporting Information files.

## Abstract

Lysosomes are acidic organelles that mediate the degradation and recycling of cellular waste materials. Damage to lysosomes can cause lysosomal membrane permeabilization (LMP) and trigger different types of cell death, including apoptosis. Newcastle disease virus (NDV) can naturally infect most birds. Additionally, it serves as a promising oncolytic virus known for its effective infection of tumor cells and induction of intensive apoptotic responses. However, the involvement of lysosomes in NDV-induced apoptosis remains poorly understood. Here, we demonstrate that NDV infection profoundly triggers LMP, leading to the translocation of cathepsin B and D and subsequent mitochondria-dependent apoptosis in various tumor and avian cells. Notably, the released cathepsin B and D exacerbate NDV-induced LMP by inducing the generation of reactive oxygen species. Additionally, we uncover that the viral Hemagglutinin neuraminidase (HN) protein induces the deglycosylation and degradation of lysosome-associated membrane protein 1 (LAMP1) and LAMP2 dependent on its sialidase activity, which finally contributes to NDV-induced LMP and cellular apoptosis. Overall, our findings elucidate the role of LMP in NDV-induced cell apoptosis and provide novel insights into the function of HN during NDV-induced LMP, which provide innovative approaches for the development of NDV-based oncolytic agents.

## Author summary

Our study investigates the role of lysosomes in Newcastle disease virus (NDV)-induced cell apoptosis. We found that NDV infection leads to lysosomal membrane permeabilization (LMP) and the release of cathepsin B and D enzymes. This triggers mitochondria-

**Funding:** This work was supported by National Natural Science Foundation of China (32202767 to Y. C); Modern Agriculture Development Special Fund of Yangzhou, Jiangsu Province, China (YZZY202303, to Y. C); Priority Academic Program Development of Jiangsu Higher Education Institutions (None, to Y. C). We confirm that the funders had no role in study design, data collection and analysis, decision to publish, or preparation of the manuscript.

**Competing interests:** The authors have declared that no competing interests exist.

dependent apoptosis in various tumor and avian cells. The released cathepsin B and D exacerbate LMP by generating reactive oxygen species. Additionally, the viral Hemagglutinin neuraminidase protein degrades lysosome-associated membrane proteins (LAMP1 and LAMP2), contributing to LMP and apoptosis. These findings provide insights into NDV-induced cell apoptosis and highlight the potential of targeting lysosomes for the development of NDV-based oncolytic therapies.

## Introduction

Lysosomes, first discovered by Christian de Duve in 1955, are single-membrane-bound and acidic vesicles. They act as the cellular "stomach," breaking down biomacromolecules into raw materials for cell growth using over 60 acid hydrolases [1]. In addition to their digestive role, lysosomes are known as "suicide bags" for cells due to their ability to induce cell death through lysosomal membrane permeabilization (LMP) [2]. Normally, lysosomes are protected by highly glycosylated transmembrane proteins like lysosome-associated membrane protein 1 (LAMP1) and LAMP2, which prevent the digestion of lysosomal membranes by acid hydrolases. However, LMP increases the permeability of the lysosomal membrane, allowing acid hydrolases to translocate to the cytosol. This massive release of lysosomal contents usually leads to uncontrolled and generalized degradation of cell components, resulting in cell necrosis [2]. However, partial and selective release of specific lysosomal enzymes, particularly cathepsins, initiates lysosome-dependent cell death (LDCD) through a cascade of cell signaling events, mainly apoptosis [3–5]. Cathepsins such as cathepsin B (CTSB) and cathepsin D (CTSD) can directly activate substrates like Bid and Bcl-2, promoting mitochondrial outer membrane permeabilization (MOMP), caspase activation, and apoptosis [2,6,7]. LDCD can also induce cell death through other programmed cell death pathways, including pyroptosis, ferroptosis and necroptosis [2]. Thus, the integrity of the lysosomal membrane plays a pivotal role in determining cell fate.

Several factors, including lysosome tropic agents, reactive oxygen species (ROS) and $Ca^{2+}$, are the main known triggers of LMP. Lysosome-targeting agents, such as LLoMe [8], CAD [9] and MSDH [10], can selectively enter the lysosomal lumen by protonation, inducing lysosomal swelling and subsequent LMP. ROS can directly penetrate the lysosomal membrane, catalyze $Fe^{2+}$ in the lysosomal lumen to undergo Fenton reaction, produce highly oxidizing hydroxyl radicals, and finally lead to LMP [11]. Specific signals can stimulate calcium channels in intracellular calcium stores, such as the endoplasmic reticulum and sarcoplasmic reticulum, causing a rapid increase in intracellular $Ca^{2+}$ concentration. This rise in $Ca^{2+}$ activates cytoplasmic calcium proteases, which degrade lysosomal membrane proteins, resulting in LMP [12]. Recent studies have identified two host proteins, phospholipase A2 group 4E protein (PLA2G4E) [13] and lysosomal cell death regulator (LCDR) [14], as regulators of LMP. PLA2G4E hydrolyzes the fatty acyl linkage of lysosomal membrane phospholipids, while LCDR reduces the stability of lysosomal-associated transmembrane protein 5 (LAPTM5) transcripts, both leading to LMP. Moreover, viral infections have been implicated in LMP induction. For instance, human immunodeficiency virus (HIV) infection of CD4+ T cells results in the transcribing and activation of the Nef protein, which, through the p53 pathway, upregulates damage-regulated autophagy modulator (DRAM). Elevated DRAM levels destabilize lysosomal membranes, triggering LMP and apoptosis [15,16]. Infections by adenovirus type 5 (Ad5) [17] and dengue virus (DENV) [18,19] induce mitochondrial stress, leading to the release ROS into the cytoplasm, resulting in LMP and apoptosis. The SARS-CoV-2 non-

structural protein 6 disrupts lysosomal function by targeting ATP6AP1, leading to NLRP3-dependent pyroptosis [20]. In summary, LMP plays a crucial role in viral-induced cell death.

Newcastle disease virus (NDV) is recognized as one of the most serious pathogens in the global poultry industry. Additionally, as an oncolytic virus, NDV has been tested as an attractive oncolytic agent for cancer virotherapy. Among various programmed cell death pathways, apoptosis is the primary mechanism by which NDV infection induces cell death and mediates pathogenicity and oncolysis [21,22]. Previous studies have demonstrated that NDV infection can induce extrinsic cell apoptosis by promoting TRAIL-mediated apoptosis [23–25]. Furthermore, NDV infection can induce mitochondrial damage, augment mitochondrial membrane potential (MMP) loss, and promote intrinsic cell apoptosis [26,27]. In addition, Li et al. reported that NDV infection can activate the eIF2α-CHOP-BCL-2/JNK and IRE1α-XBP1/JNK pathways through endoplasmic reticulum stress, resulting in cell apoptosis [23]. Despite extensive studies on NDV-induced apoptosis, the role of LMP in this process remains unclear.

In this study, we aimed to investigate whether and how NDV infection induces LMP and elucidate the role of LMP in NDV-induced apoptosis. Our results reveal that NDV relies on the sialidase activity of viral HN protein to hydrolyze the sialic acid residues at the end of the glycan chains of LAMP1 and LAMP2. This desialylation leads to deglycosylation and degradation of LAMP1 and LAMP2 by CTSB in the lysosomal lumen, subsequently triggering LMP. As a result, CTSB and CTSD translocate to the cytoplasm and mediate NDV-induced cell apoptosis by inducing MOMP. Our findings uncover a novel mechanism of NDV-induced cell apoptosis and provide valuable insights for the development of NDV-based oncolytic therapies.

## Results

### NDV infection triggers LMP, leading to the leakage of CTSB and CTSD

To investigate whether NDV infection induces LMP, we first examined its effect on lysosome acidification using Lysotracker Red, a fluorescent dye whose intensity inversely correlates with lysosomal pH. Torin 1, an mTOR inhibitor promoting lysosome biosynthesis, and bafilomycin A1 (Baf A1), a V-ATPase inhibitor inhibiting lysosomal acidification, served as positive and negative controls for Lysotracker Red staining, respectively. Both confocal microscopy (Fig 1A) and flow cytometry analysis (Fig 1B) revealed that NDV infection led to a time-dependent decrease in Lysotracker Red fluorescence in HeLa cells (human cervical cancer cells), similar to that caused by the V-ATPase inhibitor bafilomycin A1 (Baf A1, an inhibitor of lysosome acidification). Consistent results were observed in another tumor cell line, A549 cells (human lung adenocarcinoma cells) (S1A Fig), and two avian cell lines, DF-1 cells (immortal chicken embryo fibroblast) (S1B Fig) and HD11 cells (chicken macrophage) (S1C Fig), indicating that NDV infection may induce lysosomal deacidification in different tumor and avian cells.

Lysosomes are enclosed by a single membrane to maintain their acidic environment. When the membrane permeability of lysosomes increases, acidic components within the lysosomes are released into the cytoplasm, leading to lysosomal deacidification. As a result, lysosomal deacidification is considered an early indicator of LMP [2,28]. Based on this, we further assessed lysosomal membrane permeability using acridine orange (AO), a lysomotropic fluorescent dye that exhibits metachromatic properties within lysosomes. In intact lysosomes, AO exists in a protonated oligomeric form and emits red fluorescence. However, LMP causes AO to leak out, resulting in green fluorescence emitted by its deprotonated monomeric form. As shown in Fig 1C–1E, NDV infection significantly reduced red fluorescence signals and increased green fluorescence signals in HeLa cells. Similarly, AO staining in A549, DF-1, and

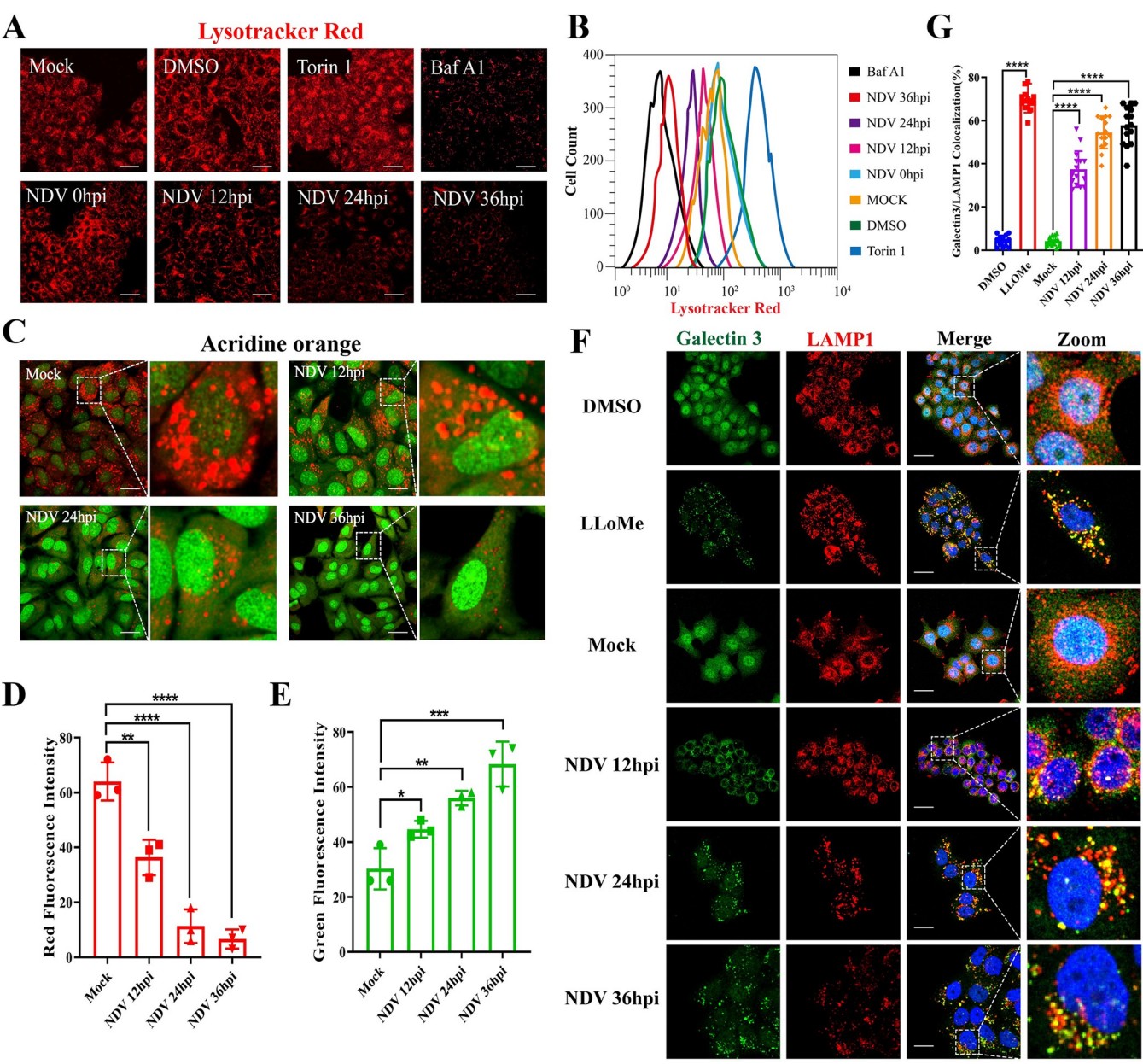

**Fig 1. NDV infection induces LMP in HeLa cells.** (A&B) HeLa cells were infected with Herts/33 at 0.01 MOI for the indicated time points. Torin 1 (1 μM, 4h) and Baf A1 (1 μM, 4h) were used as positive and negative controls, respectively. The fluorescence signals were detected by confocal microscopy (A) and flow cytometry (B) after staining the cells with Lysotracker Red (100 nM) for 1h. Scale bars, 50 μm. (C) HeLa cells were infected with Herts/33 at 0.01 MOI for the indicated time points or mock-infected. Then the AO red and green fluorescence signals were detected by confocal microscopy following incubation with AO (1 μg/mL) for 30 min. Scale bars, 20 μm. (D&E) Quantitation of AO red (D) and green (E) fluorescence signals by ImageJ software. (F) HeLa cells were infected with Herts/33 at 0.01 MOI for the indicated time points or mock-infected. LLoMe (1 mM, 4h) was used as a positive control. Then the cells were stained with rabbit anti-galectin 3 (green) and mouse anti-LAMP1 (red) antibodies and observed by confocal microscopy. Scale bars, 20 μm. (G) Manders' Colocalization Coefficients of galectin 3 with LAMP1 were measured by ImageJ software. Error bars represent SDs for triplicate analyses of >100 cells/sample (D&E), or SDs for 15 cells (G). All significance analyses were assessed using one-way ANOVA with Dunnett's multiple comparisons test.

HD11 cells also demonstrated elevated green fluorescence and decreased red fluorescence following NDV infection (S1D Fig). These results indicate that NDV infection may induce LMP in these cells.

To validate these findings, we performed a galectin puncta assay, as previously described [29]. Galectins are a family of proteins that bind to glycans containing beta-galactoside. Upon LMP, galectins translocate from the cytosol and nucleus to the lysosomal membrane. Utilizing confocal microscopy, we observed that NDV infection, similar to LLOME (an LMP inducer), caused the redistribution of galectin 3 from a diffuse distribution in the cytoplasm and nucleus to a punctate pattern, precisely colocalizing with LAMP1 (a lysosomal membrane marker) in HeLa (Fig 1F and 1G) and A549 (S1E Fig) cells. Collectively, these findings provide compelling evidence that NDV infection triggers LMP in various tumor and avian cells.

LMP often leads to the translocation of lysosomal cathepsins from the lysosomal lumen to the cytoplasm. Among these cathepsins, CTSB and CTSD are the primary active proteases after LMP [2,30–32]. To explore the effect of NDV infection on subcellular localization of CTSB and CTSD, we performed Western blotting to assess the expression of CTSB and CTSD in cell lysates extracted from the cytoplasm and lysosome fraction, respectively. In most cases, LAMP1 and LAMP2 are commonly used as lysosomal markers due to their abundant expression on the lysosomal membrane [33]. However, we observed a gradual decrease in the expression levels of LAMP1 and LAMP2 following NDV infection in HeLa cells, while the expression level of another lysosomal marker protein, LAMP3 [33], remained stable (Fig 2A). Therefore, in our study, we selected LAMP3 as a lysosomal marker for the quantitative analysis of lysosomal abundance. Additionally, the composition of the cytoplasmic fraction was confirmed by the α-tubulin protein. As shown in Fig 2A, the expression levels of CTSB and CTSD exhibited time-dependent decreases in the lysosomes (Fig 2B and 2D), while showing time-dependent increases in the cytoplasm (Fig 2C and 2E). Furthermore, we measured the activity of CTSB and CTSD, which demonstrated a significant increase in both cathepsins' activity in the cytoplasmic fraction extracted from NDV-infected HeLa cells compared to mock-infected cells (Fig 2F and 2G). Additionally, we utilized confocal microscopy to assess the colocalization between CTSB or CTSD and LAMP3 at different timepoints after NDV infection. The results supported the gradual diffusion of CTSB (Fig 2H and 2I) or CTSD (Fig 2K and 2J) throughout the cytosol upon NDV infection, resulting in a significant decrease in the colocalization with LAMP3. This gradual dispersal of CTSB and CTSD with infection time was also observed in NDV-infected A549 cells (S2 Fig). In conclusion, these results suggest that NDV infection triggers LMP in different tumor and avian cells, leading to the leakage of CTSB and CTSD from the lysosomal lumen to the cytoplasm.

## LMP promotes NDV-induced apoptosis by inducing mitochondrial dysfunction via CTSB and CTSD

The leaked lysosomal cathepsins have been implicated in the regulation of various cell death pathways, including apoptosis, necrosis, ferroptosis, and pyroptosis. Given that apoptosis is a major hallmark of NDV-mediated cytotoxicity [21,22], we focused our research on the role of LMP in NDV-induced apoptosis. To this end, we selected HeLa and DF-1 cells as representative models of tumor and avian cells. These cells were first infected by Herts/33 for 24h, and then treated with CA-074 (CTSB specific inhibitor) and Pepstatin A (Pep A, an aspartic acid protease inhibitor) for an additional 6h. We found that treatment of CA-074 and Pep A showed no significant effect on viral NP protein levels (Fig 3D and 3G) but significantly suppressed NDV-induced apoptosis in Hela cells (Fig 3B and 3C) and DF-1 cells (Fig 3E and 3F). These results indicate that LMP promotes NDV-induced apoptosis.

Caspases are a family of genes that play a crucial role in maintaining homeostasis by regulating cell apoptosis and inflammation. Caspases involved in apoptosis can be categorized into initiator caspases (caspase 8 and 9) and executioner caspases (caspase 3, 6, and 7), based on

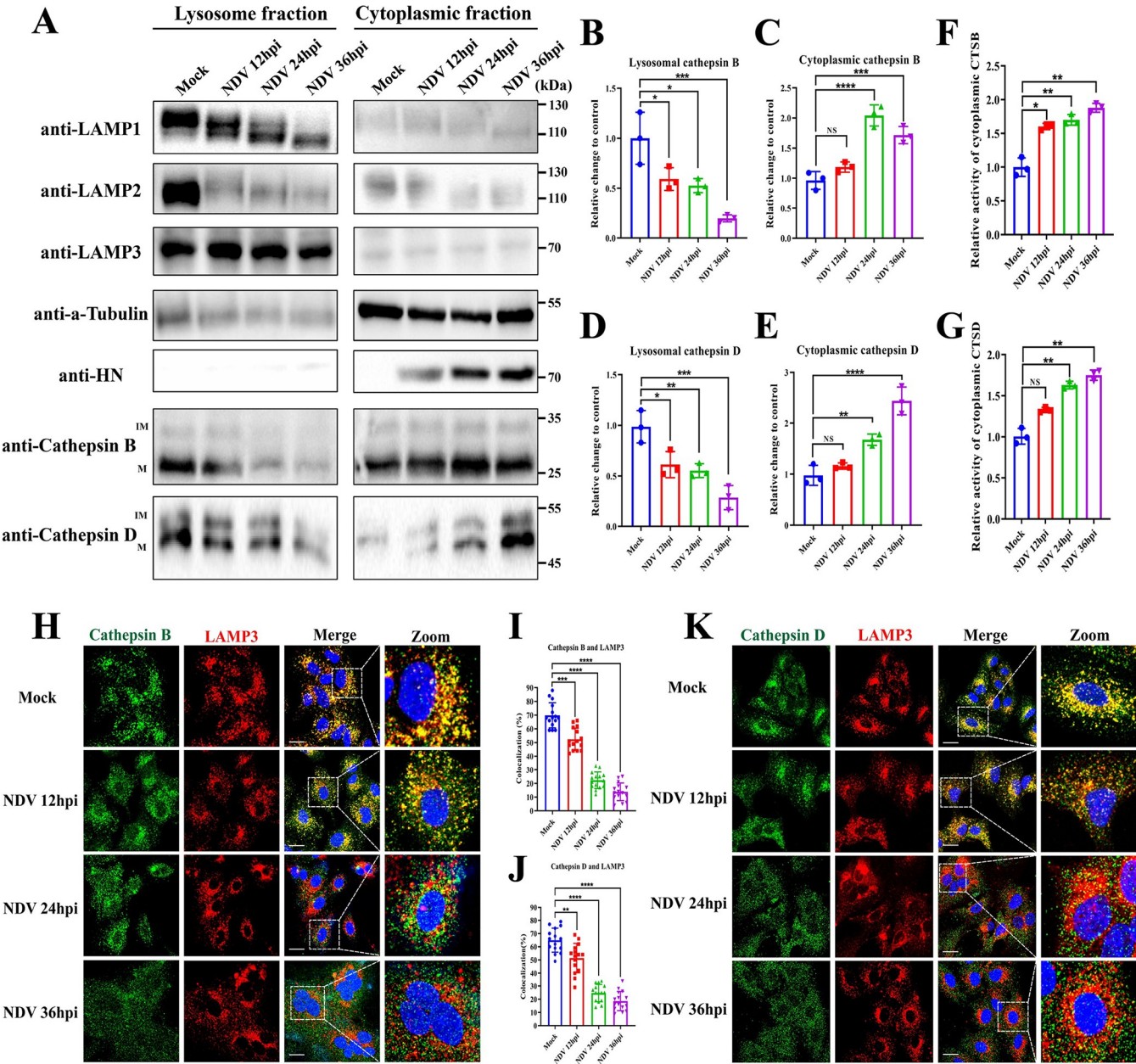

**Fig 2. NDV infection induces the leakage of CTSB and CTSD from the lysosomal lumen to the cytoplasm in HeLa cells.** (A) HeLa cells infected with Herts/33 at 0.01 MOI for the indicated time points or mock-infected. Indicated proteins in both lysosomal and cytoplasmic fractions extracted from the cells were detected by Western blotting. Immature and mature types of cathepsins were indicated as "IM" and "M", respectively. (B-E) The relative intensity of lysosomal CTSB (B) and lysosomal CTSD (D) or cytoplasmic CTSB (C) and cytoplasmic CTSD (E) were quantified by ImageJ software. LAMP3 and α-Tubulin were used as normalized controls for lysosomal and cytoplasmic fractions, respectively. (F&G) HeLa cells were infected with Herts/33 at 0.01 MOI for the indicated time points or mock-infected. The activity of cytoplasmic CTSB (F) and CTSD (G) was measured as described in Materials and Methods. The obtained fluorescence intensity was normalized to the concentration of individual protein. (H&K) HeLa cells were infected with Herts/33 at 0.01 MOI for indicated timepoints or mock-infected. The colocalizations of CTSB (H) or CTSD (K) with LAMP3 were observed using confocal microscopy after immunostaining the cells with indicated antibodies. Scale bars, 20 μm. (I&J) Manders' Colocalization Coefficients of CTSB (I) or CTSD (J) with LAMP3 were quantified by ImageJ software. Error bars represent SDs for triplicate analyses of three independent experiments (B-F) or SDs for 15 cells (I&J). All significance analyses were assessed using one-way ANOVA with Dunnett's multiple comparisons test.

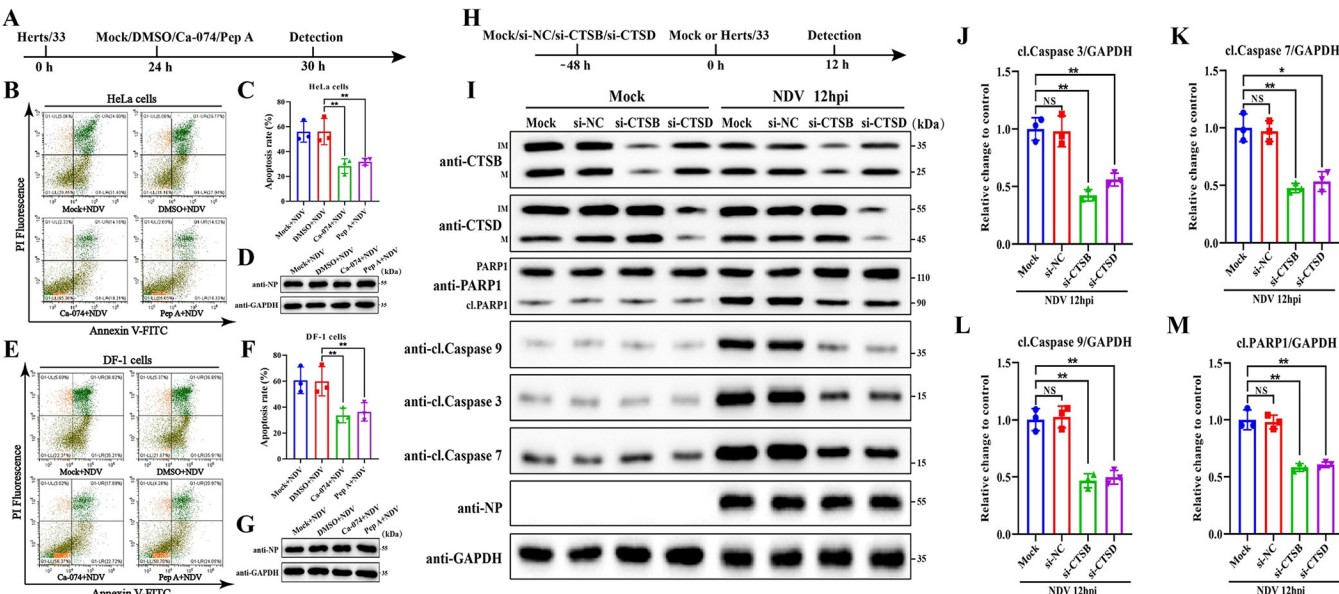

**Fig 3. LMP promotes NDV-induced apoptosis through the involvement of CTSB and CTSD.** (A) Schematic diagram of the experimental design for (B-G). HeLa cells or DF-1 cells were infected with Herts/33 at 0.01 MOI for 24h. Then, the cells were treated with Ca-074 (20 μM), Pep A (20 μM), DMSO, or mock-treated for 6h. (B&C&E&F) The apoptosis level of HeLa cells (B) or DF-1 cells (E) was measured by AnnexinV-FITC/PI staining using flow cytometry, and quantitation of the apoptosis rate of HeLa cells (C) and DF-1 cells (F) are shown, respectively. (D&G) The NP protein in HeLa cells (D) or DF-1 cells (G) was detected by Western blotting. (H) Schematic diagram of the experimental design for (I-M). HeLa cells were transfected with si-CTSB, si-CTSD, si-NC, or mock-transfected for 48h. Then, the cells were infected with Herts/33 at 0.01 MOI or mock-infected for 12h. (I-M) The indicated proteins were detected by Western blotting (I). The relative intensity of cleaved caspase 3 (J), cleaved caspase 7 (K), cleaved caspase 9 (L), and cleaved PARP1 (M) in NDV-infected cells were quantified by ImageJ software. GAPDH was used as a normalized control. All error bars represent SDs for triplicate analyses of three independent experiments. All significance analyses were assessed using one-way ANOVA with Dunnett's multiple comparisons test.

their mechanism of action [34]. To further confirm the LMP-mediated pro-apoptotic role during NDV infection, we examined the impact of small interfering RNA (siRNA) targeting CTSB (si-CTSB) and CTSB (si-CTSD) on the expression levels of caspase 3, caspase 7, and caspase 9 in HeLa cells during NDV infection using Western blotting. As shown in Fig 3I, knockdown of CTSB or CTSD showed no significant effect on viral NP protein but significantly inhibited NDV-induced apoptosis, as evidenced by the decrease of cleaved caspase 3 (Fig 3J), cleaved caspase 7 (Fig 3K), cleaved caspase 9 (Fig 3L), and the cleaved fragment of caspase 3 substrate poly [ADP-ribose] polymerase 1 (PARP1) (Fig 3M). These findings strongly indicate that LMP promotes NDV-induced apoptosis through the involvement of CTSB and CTSD.

Following LMP, the translocated cathepsins often contribute to mitochondrial damage, leading to apoptosis through the mitochondrial pathway. This process involves the release of cytochrome c, activation of caspases, and subsequent cell apoptosis [35,36]. In line with these observations, knockdown of CTSB and CTSD significantly inhibited the cleavage of caspase 3, caspase 7, and caspase 9 after NDV infection, which are recognized as key effectors of mitochondria-dependent apoptosis (Fig 3I). Given the above findings, we aimed to investigate whether LMP promotes NDV-induced apoptosis via the mitochondrial pathway. We initially examined the role of LMP in mitochondrial damage during NDV infection by assessing MMP using JC-1 dye. According to flow cytometry analysis (Fig 4A and 4C) and fluorescence microscope (Fig 4B and 4D) results, NDV infection alone led to a significant reduction in MMP, as indicated by the decrease in the ratio of JC-1 aggregates (Red) to monomers (Green), while knockdown of CTSB and CTSD significantly restored this loss of MMP, respectively. These results indicate LMP promotes mitochondrial damage during NDV infection through the involvement of CTSB and CTSD.

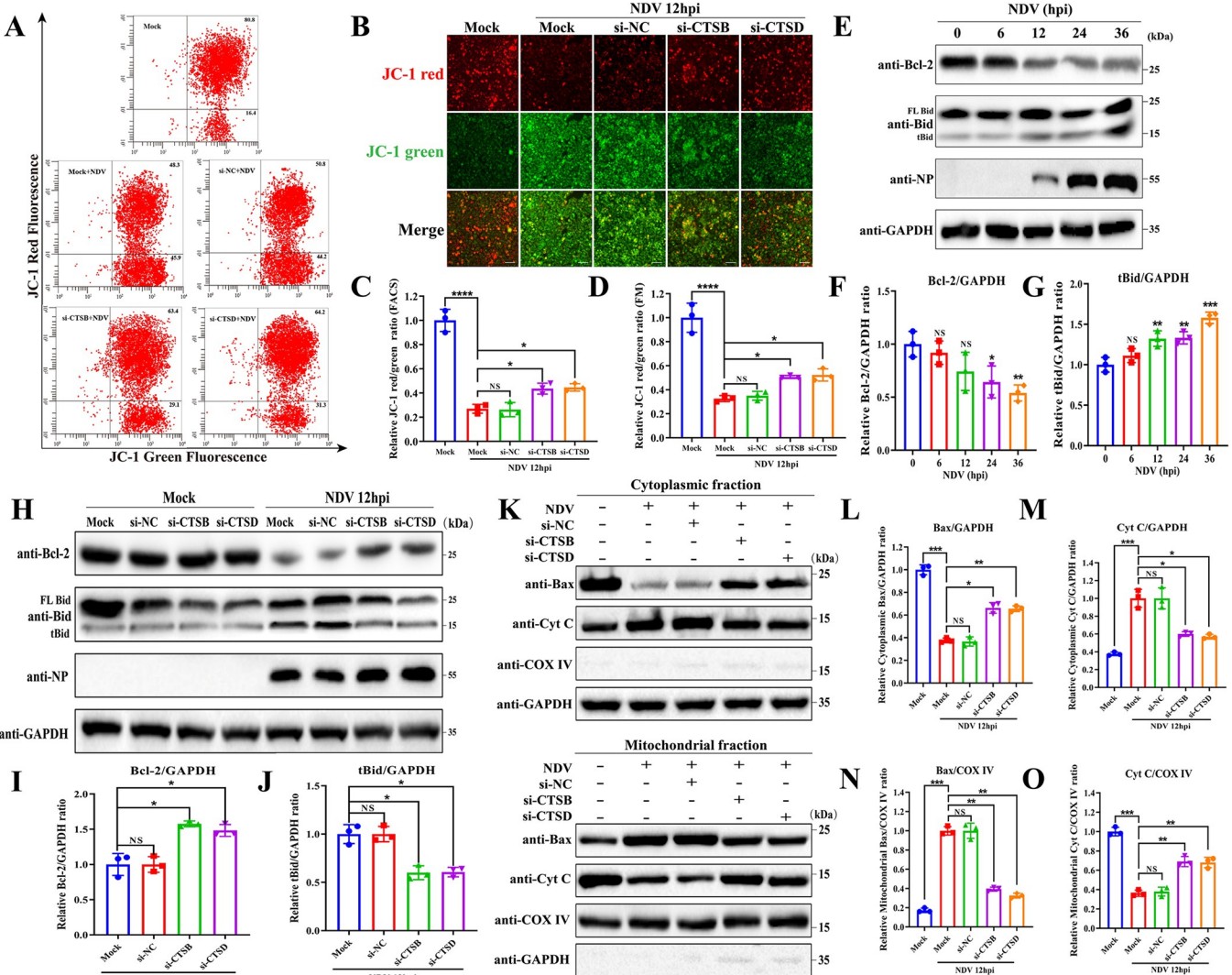

**Fig 4. LMP promotes NDV-induced apoptosis by inducing mitochondrial dysfunction via CTSB and CTSD.** (A&B) HeLa cells were transfected with si-CTSB, si-CTSD, or si-NC for 48h or mock-transfected. The MMP level was detected by flow cytometry (A) or fluorescence microscope (B) using JC-1 probe staining following infection with Herts/33 at 0.01 MOI for 12h or mock-infected. Scale bars, 100 μm. (C&D) Quantitation of JC-1 red to green fluorescence ratio detected by flow cytometry (C) and fluorescence microscope (D) by ImageJ software are shown, respectively. (E-G) Indicated proteins were detected by Western blotting in HeLa cells infected with Herts/33 at 0.01 MOI for indicated timepoints (E). The relative intensity of Bcl-2 (F) and tBid (G) were quantified by ImageJ software. GAPDH was used as a normalized control. (H-J) HeLa cells were transfected with si-CTSB, si-CTSD, or si-NC for 48h or mock-transfected. Subsequently, the indicated proteins were detected by Western blotting following infection with Herts/33 at 0.01 MOI for 12h (H). The relative intensity of Bcl-2 (I) and tBid (J) in NDV-infected cells were quantified by ImageJ software. GAPDH was utilized as a normalized control. (K-O) HeLa cells were transfected with si-CTSB or si-CTSD, or si-NC for 48h or mock-transfected. The expression levels of Bax and Cyt C in the cytoplasmic and mitochondrial fractions were determined by Western blotting after infection with Herts/33 at 0.01 MOI or mock-infection for 12h (K). The relative intensity ratio of Bax/GAPDH (L), Cyc/GAPDH (M), Bax/COX IV (N), and Cyc/COX IV (O) were quantified by ImageJ software, respectively. GAPDH was used as a cytoplasmic fraction control and COX IV was used as a mitochondrial fraction control. All error bars represent SDs for triplicate analyses of three independent experiments. All significance analyses were assessed using one-way ANOVA with Dunnett's multiple comparisons test.

MMP and mitochondria-dependent apoptosis are governed by intracellular proteins known as Bcl-2 family, including pro-apoptotic effectors (e.g., Bax), pro-apoptotic sensors (e.g., Bid), and anti-apoptotic members (e.g., Bcl-2) [37]. Thus, we further detected the impact of LMP on the expression of these proteins. We found that NDV infection led to a time-dependent decrease of Bcl-2 protein (Fig 4E and 4F) and a time-dependent increase in tBid (Fig 4E

and 4G), indicating the induction of mitochondria-dependent apoptosis. Furthermore, si-CTSB and si-CTSD significantly suppressed mitochondria-dependent apoptosis following NDV infection, as demonstrated by the increase in Bcl-2 (Fig 4H and 4I), and the decrease in tBid (Fig 4H and 4J). Additionally, NDV infection promoted the translocation of Bax from the cytoplasm to mitochondria and facilitated the release of cytochrome C (Cyt C) from the mitochondria to the cytoplasm, whereas knockdown of CTSB and CTSD significantly prevented these processes (Fig 4K–4O). Collectively, these results suggest that LMP relies on CTSB and CTSD to mediate mitochondrial dysfunction and induce mitochondrial pathway-dependent apoptosis, ultimately promoting NDV-induced cell apoptosis.

The impact of LMP on NDV replication was also evaluated in HeLa and DF-1 cells as well. LMP was induced using LLoMe, while suppressed by inhibiting the activity of CTSB and CTSD using Ca-074 and Pep A, respectively. Viral growth curves and the expression levels of viral proteins were compared between these chemicals-treated cells and mock-treated cells. The results indicated that LLoMe-treated cells exhibited a significantly increased viral titer in the culture supernatant, while Ca-074 or Pep A-treated cells showed reduced titers compared to mock-treated cells (S3A and S3B Fig). Notably, LLoMe displayed a more pronounced ability to promote NDV replication in the tumor HeLa cells compared to the avian DF-1 cells, especially at 12 and 24 hours post-infection (hpi). This discrepancy may be attributed to the greater fragility and susceptibility of tumor cell lysosomes to LMP as compared to normal cells [38,39]. Western blotting analysis further confirmed a notable increase in the levels of viral NP and HN proteins in LLoMe-treated cells, while Ca-074 or Pep A-treated cells showed decreased protein levels compared to the mock-treated cells (S3C and S3D Fig). These findings suggest that LMP facilitates NDV replication in tumor and avian cells.

## CTSB and CTSD exacerbate NDV-induced LMP by inducing the generation of ROS

Mitochondrial dysfunction disrupts the balance between ROS production and antioxidant defense, leading to an increased release of ROS [40,41]. In contrast to mitochondria, lysosomes lack typical antioxidant enzymes, making their membrane susceptible to damage under conditions of high ROS levels [11,42,43]. Given the crucial role of ROS in maintaining lysosomal membrane integrity and the aforementioned findings indicating that NDV-induced LMP promotes mitochondrial dysfunction via CTSB and CTSD, we hypothesized that CTSB and CTSD further exacerbate NDV-induced LMP by inducing ROS generation during NDV infection. To verify this hypothesis, we first measured the levels of ROS during NDV infection by flow cytometry analysis. Our results revealed a significant time-dependent increase in ROS levels upon NDV infection (Fig 5A). Moreover, treatment with the antioxidant N-acetyl-L-cysteine (NAC) significantly reduced the extent of LMP following NDV infection (Fig 5D), along with a decrease in ROS levels (S4 Fig). These results demonstrated that NDV infection induces the generation of ROS, which in turn promotes NDV-induced LMP. Furthermore, we explored whether LMP contributes to the increased generation of ROS during NDV infection. Our results showed that knockdown of either CTSB or CTSD significantly decreased the levels of ROS (Fig 5B), while treatment with LLoMe promoted ROS generation following NDV infection (Fig 5C), suggesting that LMP contributes to the generation of ROS during NDV infection. Taken together, these findings demonstrate a vicious cycle of NDV-induced lysosomal and mitochondrial damage, in which CTSB and CTSD act as effectors of LMP and further aggravate NDV-induced LMP by inducing mitochondrial dysfunction and increasing ROS production and release.

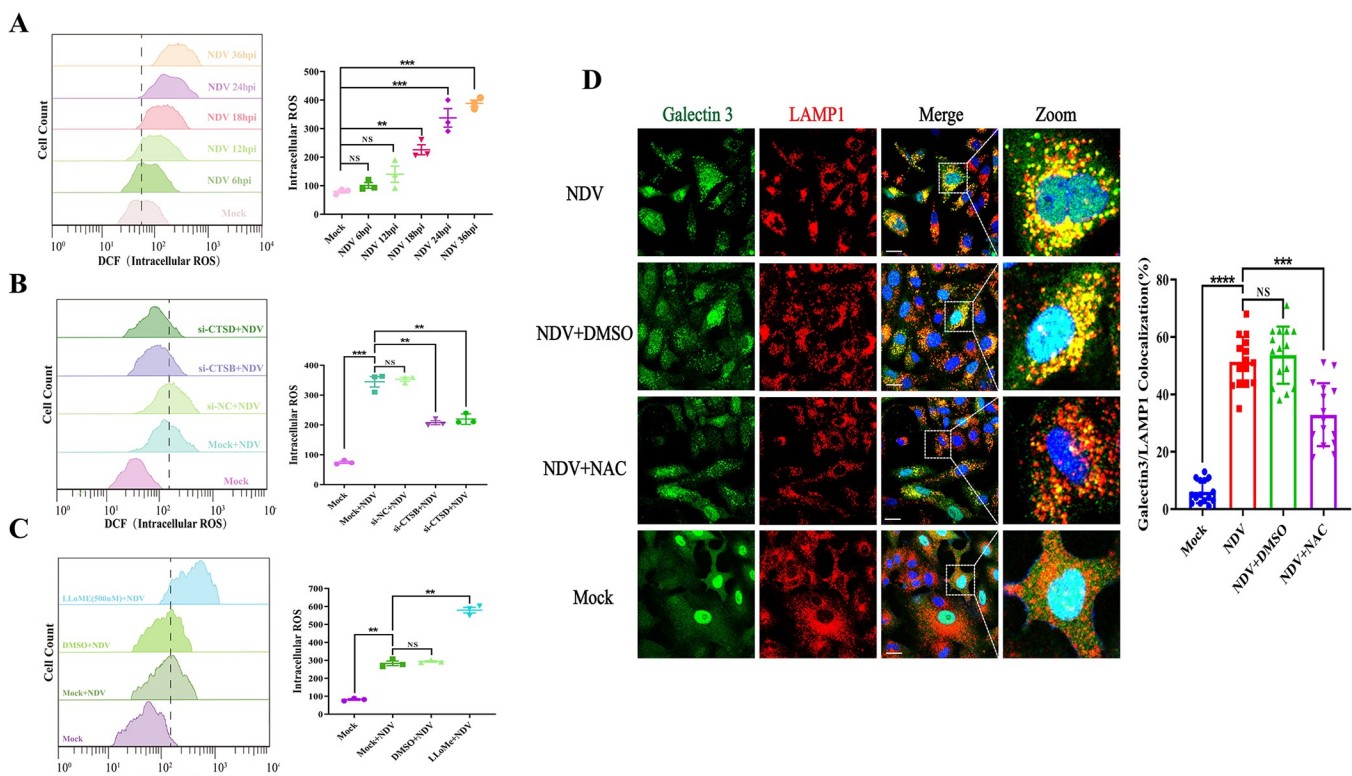

**Fig 5. CTSB and CTSD exacerbate NDV-induced LMP by inducing the generation of ROS.** (A) Intracellular ROS levels were measured by flow cytometry using DCFH-DA staining in HeLa cells infected with Herts/33 at 0.01 MOI for different timepoints or mock-infected. (B) HeLa cells were transfected with si-CTSB, si-CTSD, or si-NC for 48h, or mock-transfected, and intracellular ROS levels were measured by flow cytometry using DCFH-DA staining following infection with Herts/33 at 0.01 MOI or mock-infection for 12h. (C) HeLa cells were pretreated with LLoMe (500 nM), DMSO, or mock-untreated for 3h, respectively. Subsequently, the intracellular ROS levels were measured by flow cytometry using DCFH-DA staining cells following infection with Herts/33 at 0.01 MOI or mock-infection for 24h. (D) HeLa cells were treated with NAC (2 mM) for 24h after absorption with Herts/33 at 0.01 MOI for 1h. LMP levels were evaluated by confocal microscopy using indicated antibodies. Scale bars, 20 μm. Manders' Colocalization Coefficients of galectin 3 with LAMP1 were quantified by ImageJ software and shown on the right. Error bars represent SDs for triplicate analyses of three independent experiments (A-C) or SDs for 15 cells (D). All significance analyses were assessed using one-way ANOVA with Dunnett's multiple comparisons test.

## NDV infection induces deglycosylation and degradation of LAMP1 and LAMP2

After establishing the role of LMP in NDV-induced cell apoptosis, it is necessary to explore the mechanism by which NDV induces LMP. LAMP1 and LAMP2 are crucial constituents of the lysosomal membrane, maintaining its integrity by protecting it from lysosomal enzymes through their highly glycosylated luminal layer in the lysosomal lumen [44]. Despite no significant difference in the mRNA levels of LAMP1 and LAMP2 (Fig 6A), immunofluorescence analysis revealed a significant decrease in the intensity of LAMP1 and LAMP2 following NDV infection (Fig 6B). Consistently, Western blotting analysis also demonstrated a time- and dose-dependent reduction in the protein levels of LAMP1 and LAMP2 in HeLa cells following NDV infection (Fig 6C). Normally, LAMP1 and LAMP2 exhibit molecular weights of approximately 120–130 kDa due to glycosylation, but after NDV infection, their molecular weights gradually decreased to around 70 kDa, indicating deglycosylation of these proteins in HeLa cells (Fig 6C). Moreover, a digestion assay using the glycosidase PNGase F demonstrated that, following NDV infection, in addition to protein deglycosylation, there was also a decrease in the expression levels of the proteins themselves, indicating the involvement of deglycosylation in the reduction of LAMP1 and LAMP2 (Fig 6D).

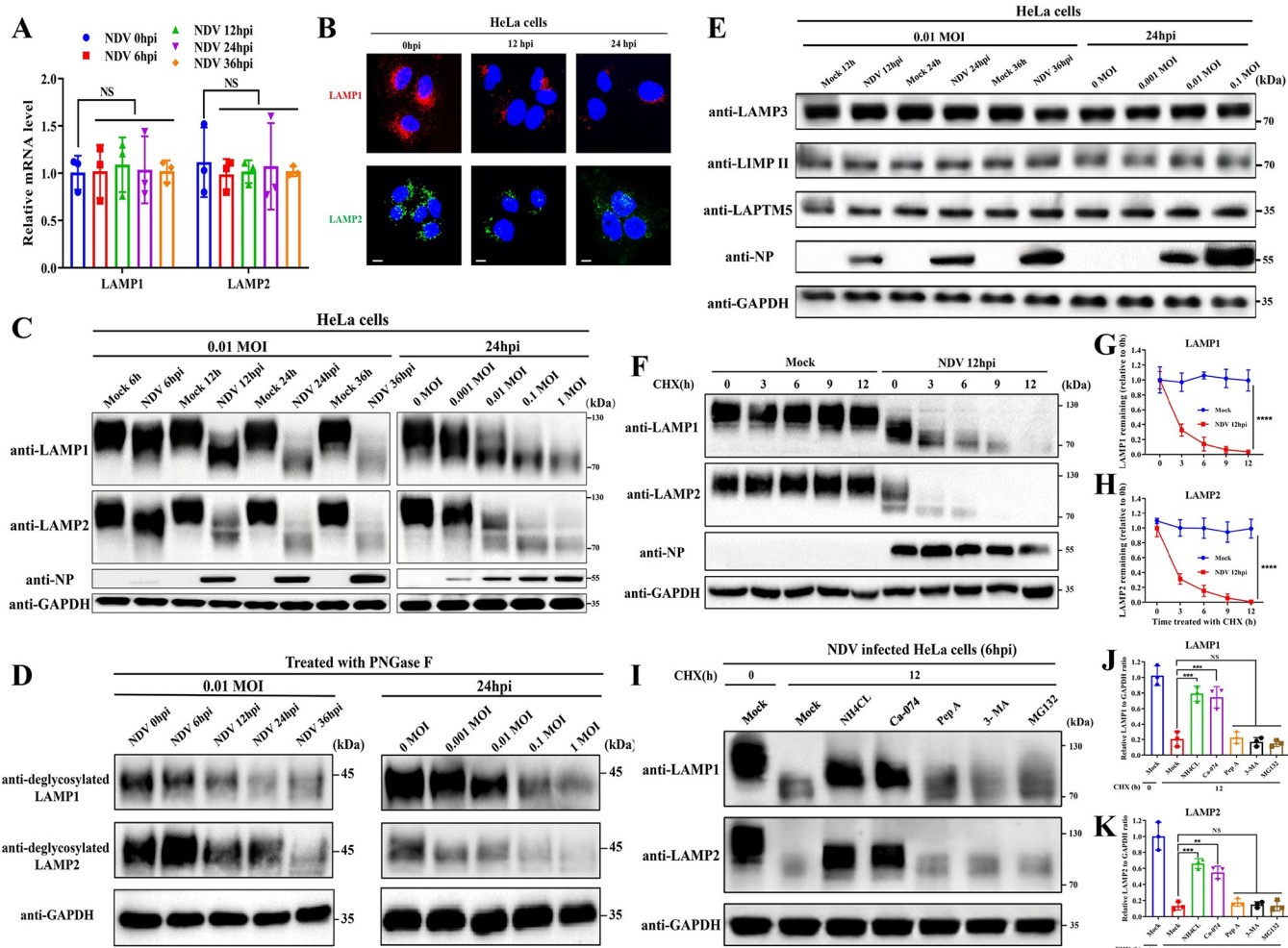

**Fig 6. NDV infection induces deglycosylation and degradation of LAMP1 and LAMP2.** (A&B) HeLa cells infected with Herts/33 at 0.01 MOI for indicated timepoints. The mRNA levels (A) and fluorescence images (B) of LAMP1 and LAMP2 were measured by RT-qPCR or observed by confocal microscopy, respectively. The mRNA levels of LAMP1 and LAMP2 were normalized to GAPDH using the $2^{-\Delta\Delta Ct}$ method. Scale bars, 20 μm. (C-E) HeLa cells infected with Herts/33 at 0.01 MOI for indicated time points or infected with Herts/33 at indicated MOIs for 24h. The protein levels of LAMP1 and LAMP2 (C) or LAMP3, LIMPII, and LAPTM5 (E) were assessed by Western blotting. The deglycosylated LAMP1 and LAMP2 in PNGase F-treated cell lysates were analyzed by Western blotting (E). (F-H) HeLa cells were infected with Herts/33 at 0.01 MOI for 12h or mock-infected. The remaining amounts of LAMP1 and LAMP2 after treatment with CHX (20 μg/mL) for indicated time points were measured using Western blotting (F). The remaining amount of LAMP1 (G) and LAMP2 (H) was calculated as the fold change of the protein present at 0h. (I-K) NDV-infected (0.01 MOI, 6h) HeLa cells were co-treated with CHX (20 μg/mL) and NH4CL (50 mM), or Ca-074 (10 μM), or Pep A (10 μM), or 3-MA (5mM), or MG132 (5 μM) or mock-treated for 12h. LAMP1 and LAMP2 protein levels in cell lysates were analyzed by Western blotting (I). The remaining amounts of LAMP1 (J) and LAMP2 (K) were calculated as the fold change of the protein to mock-treated cells at 0h. All quantitative data represent means ± SD (n = 3). Significance was assessed using Two-way ANOVA with Dunnett's multiple comparisons test (A), or unpaired two-tailed Student's t-test (G&H), or One-way ANOVA with Dunnett's multiple comparisons test (J&K).

We further examined the expression of three other lysosomal membrane proteins, including two glycosylated proteins, LAMP3 and lysosomal integral membrane protein-2 (LIMP II), and a non-glycosylated protein, lysosomal-associated protein transmembrane 5 (LAPTM5). Unexpectedly, NDV infection showed no significant effect on both mRNA and protein levels of any of these proteins in HeLa cells (Figs 6E and S5D). NDV-mediated reduction and deglycosylation of LAMP1 and LAMP2 proteins without affecting the expression of other lysosomal membrane proteins were also observed in A549 (Figs 5E and S5A) and DF-1 cells (Figs 5C, 5F, and S5B).

Since NDV infection did not affect LAMP1 and LAMP2 mRNA levels, we investigated whether the reduction in LAMP1 and LAMP2 is due to post-transcriptional regulation. To

address this, we compared the protein levels of LAMP1 and LAMP2 in NDV-infected and mock-infected HeLa cells following treatment with the translation inhibitor cycloheximide (CHX). As demonstrated in Fig 6F–6H, a significant decrease in LAMP1 and LAMP2 proteins was observed in CHX-treated cells compared to mock-infected cells following NDV infection, indicating an increase in post-transcriptional protein degradation of LAMP1 and LAMP2. To characterize the potential mechanism of LAMP1 and LAMP2 protein degradation, we measured the decrease in LAMP1 and LAMP2 protein in NDV-infected HeLa cells treated with CHX in combination with the lysosome acidification inhibitor NH4CL, CTSB inhibitor CA-074, aspartic acid protease inhibitor Pep A, autophagy inhibitor 3-Methyladenine (3-MA), or proteasome inhibitor MG132. The results showed that treatment with NH4CL or CA-074 significantly inhibited the degradation of both LAMP1 and LAMP2 compared to the cells only treated with CHX, whereas Pep A, 3-MA, and MG132 exhibited minimal effect (Fig 6I–6K). This result indicates that NDV-induced degradation of LAMP1 and LAMP2 is dependent on lysosomal acidification and CTSB activity. Together, these findings demonstrate that NDV infection leads to the deglycosylation and subsequent degradation of LAMP1 and LAMP2. This process occurs through a post-transcriptional mechanism and is dependent on lysosomal acidification and CTSB activity.

## The viral HN protein contributes to NDV-induced LMP and apoptosis

To elucidate the molecular basis of NDV-induced deglycosylation of LAMP1 and LAMP2, we transfected plasmids expressing viral structural proteins (NP, P, M, F, HN, L protein) or non-structural proteins (V and W protein) into HeLa cells and determined their effects on the expression of LAMP1 and LAMP2, respectively. As demonstrated in Fig 7A, only transfection with HN protein resulted in the deglycosylation and degradation of LAMP1 and LAMP2. Furthermore, we observed that the deglycosylation and degradation of LAMP1 and LAMP2 induced by the HN protein were dose-dependent (Fig 7B), whereas the F protein, used as a negative control, had no significant effect (Fig 7C). Importantly, transfection of the HN protein plasmid induced LMP in HeLa cells, as demonstrated by the galectin puncta assay, whereas the F protein showed no effect (Fig 7D). Co-immunoprecipitation and confocal microscopy assays showed that HN protein interacts with both LAMP1 (Fig 7E and 7G) and LAMP2 (Fig 7F and 7H) within cells. In addition, we also observed that transfection with the HN protein alone promoted cell apoptosis (S6 Fig), consistent with previous studies [45–47], highlighting the crucial role of the HN protein in NDV-induced apoptosis.

To confirm whether HN protein-induced LMP and apoptosis depend on the deglycosylation and degradation of LAMP1 and LAMP2, we overexpressed or knocked down LAMP1 and LAMP2 and assessed LMP and apoptosis rate following transfection with the HN protein. The efficiency of overexpression and knockdown was confirmed by Western blotting (Fig 7K). Galectin 3 puncta and Annexin V-FITC/PI assays revealed that LMP (Fig 7I and 7L) and apoptosis rate (Fig 7N) were suppressed when LAMP1 and LAMP2 were overexpressed, whereas knockdown of these proteins increased LMP formation (Fig 7J and 7M) and promoted cell apoptosis (Fig 7O). These results collectively support the notion that the viral HN protein plays a crucial role in NDV-induced LMP and apoptosis by directly interacting with LAMP1 and LAMP2, leading to their deglycosylation and degradation.

## The sialidase activity of the HN protein mediates the deglycosylation and degradation of LAMP1 and LAMP2

Sialic acid is the only monosaccharide involved in protein glycosylation modifications that carries a negative charge. It is commonly located at the terminus of sugar chains and plays a crucial role in maintaining the conformation of glycoproteins [48]. Proper sialylation of LAMPs

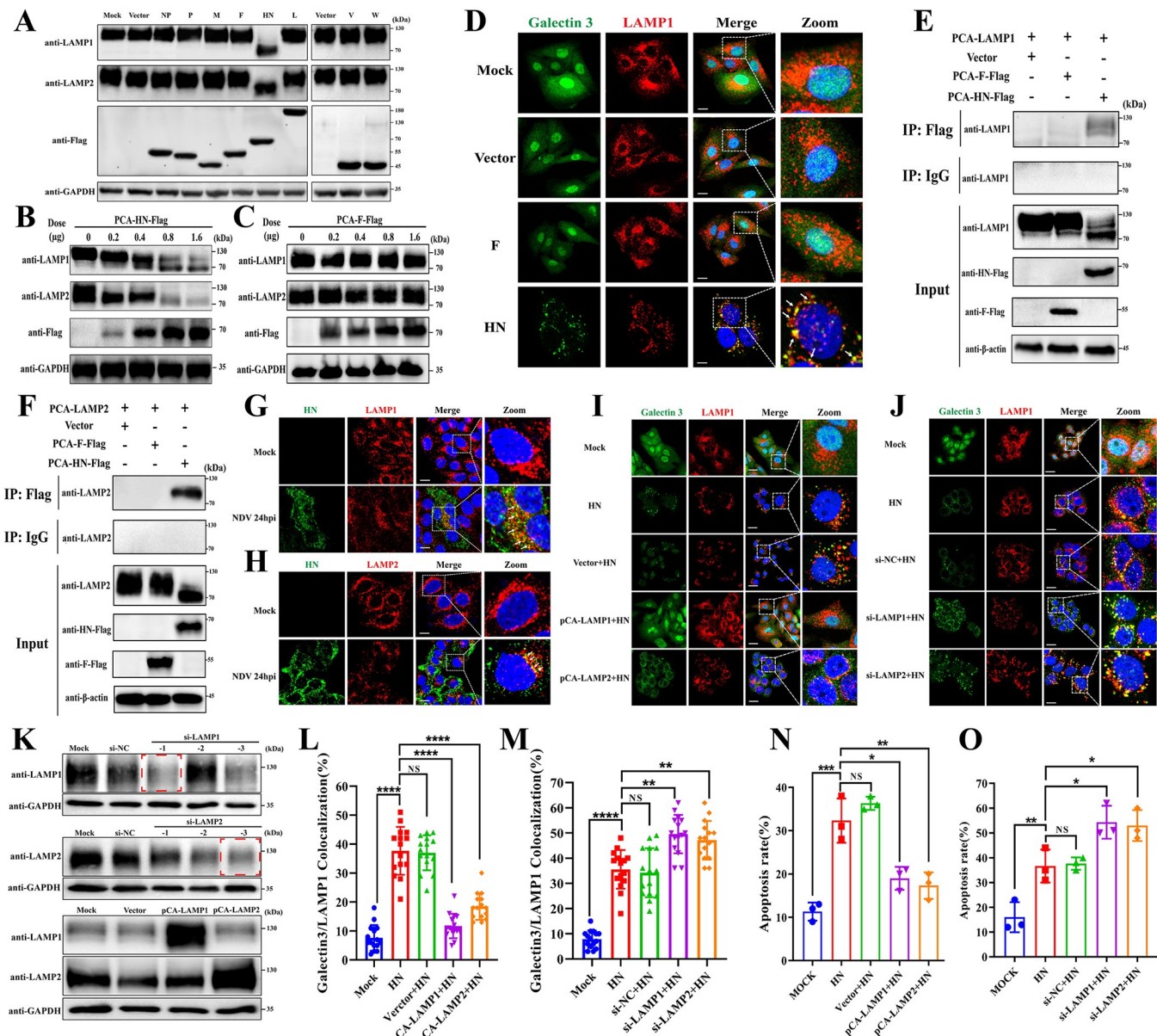

**Fig 7. The viral HN protein contributes to NDV-induced LMP and apoptosis.** (A) Western blotting analysis of LAMP1 and LAMP2 protein levels in HeLa cells after transfection with plasmids expressing indicated viral proteins for 48h. (B&C) Western blotting analysis of LAMP1 and LAMP2 protein levels in HeLa cells after transfection with indicated doses of plasmids expressing HN (B) or F (C) protein for 48h. (D) Confocal microscopy images of LMP level in HeLa cells using anti-galectin 3 (green) and anti-LAMP1 (red) antibodies after transfection with indicated plasmids for 48h. Scale bars, 20 μm. (E&F) The interaction between HN and LAMP1 (E) or LAMP2 (F) was detected by an immunoprecipitation assay with anti-flag or control IgG antibodies after transfection with indicated plasmids into HeLa cells for 48h. (G&H) Confocal microscopy images of the interaction between HN and LAMP1 (G) or LAMP2 (H) using anti-HN, anti-LAMP1, or anti-LAMP2 antibodies after infection of HeLa cells with Herts/33 at 0.01 MOI or mock-infected for 24h. Scale bars, 20 μm. (K) HeLa cells were transfected with siRNAs targeting LAMP1 or LAMP2 or plasmids expressing LAMP1 or LAMP2 for 48h. Then LAMP1 or LAMP2 protein levels were detected by Western blotting. The red boxes indicate the siRNA selected for this study. (I&J) HeLa cells were transfected with plasmids expressing LAMP1 or LAMP2 (M) or siRNAs targeting LAMP1 or LAMP2 (N) for 48h. Then the LMP level was observed by confocal microscopy using anti-galectin 3 (green) and anti-LAMP1 (red) antibodies following transfection with plasmid expressing HN protein for another 48h. Scale bars, 20 μm. (L&M) Manders' Colocalization Coefficients of galectin 3 with LAMP1 are quantified by ImageJ software and shown in (L) for (I) and (M) for (J). (N&O) HeLa cells were transfected with plasmids expressing LAMP1 or LAMP2 (N) or siRNAs targeting LAMP1 or LAMP2 (O) for 48h. Then the apoptosis rate was measured by AnnexinV-FITC/PI staining using flow cytometry following transfection with plasmid expressing HN protein for 48h. Error bars are SDs for 15 cells (L&M), or SDs for a triplicate analysis of three independent experiments (N&O). All significance analyses were assessed using One-way ANOVA with Dunnett's multiple comparisons test.

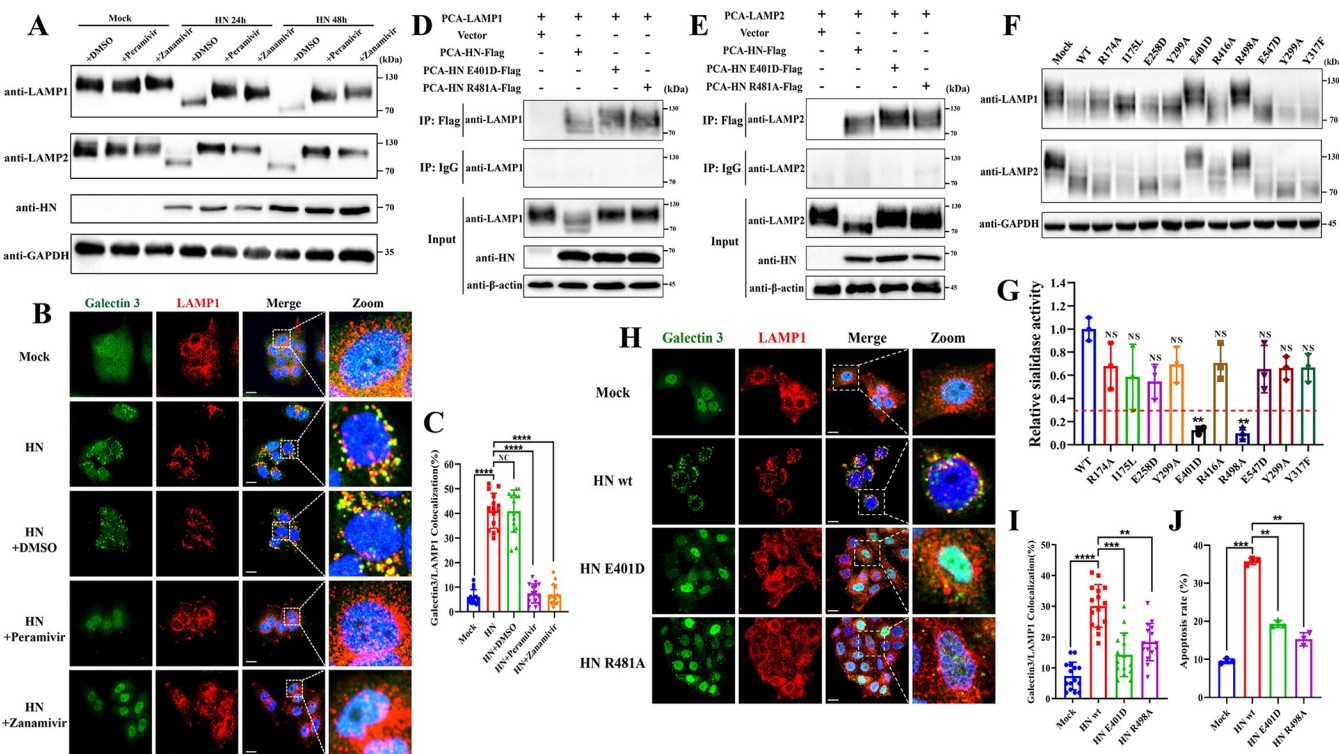

**Fig 8. The sialidase activity of the HN protein mediates the deglycosylation and degradation of LAMP1 and LAMP2.** (A) Western blotting analysis of LAMP1 and LAMP2 deglycosylation and degradation in HeLa cells transfected with plasmid expressing HN protein and treated with Peramivir (30 μg/mL) or Zanamivir (20 μg/mL) at the indicated time points post-transfection. (B&C) HeLa cells were transfected with plasmids expressing HN protein or mock-transfected. Then the cells were treated with Peramivir (30 μg/mL) or Zanamivir (20 μg/mL) or DMSO or untreated for 48h. The LMP level was observed by confocal microscopy using anti-galectin 3 (green) and anti-LAMP1 (red) antibodies. Scale bars, 20 μm (B). Manders' Colocalization Coefficients of galectin 3 with LAMP1 are quantified by ImageJ software and shown in (C). (G) Sialidase activity was measured in HeLa cells transfected with the indicated plasmids for 48h and is presented as relative change to the cells transfected with plasmids expressing wild-type HN. The red dotted line indicates the difference threshold. (F) Western blotting analysis of LAMP1 and LAMP2 deglycosylation and degradation in HeLa cells transfected with the indicated plasmids or mock-transfected for 48h. (D&E) The interaction between HN and LAMP1 (D) or LAMP2 (E) in HeLa cells was detected by an immunoprecipitation assay with anti-flag or control IgG antibodies after transfection with indicated plasmids for 48h. (H) The LMP level was observed by confocal microscopy using anti-galectin 3 (green) and anti-LAMP1 (red) antibodies following transfection with the indicated plasmids or mock-transfected for 48h. Scale bars, 20 μm. (I) Manders' Colocalization Coefficients of galectin 3 with LAMP1 are calculated by ImageJ software. (J) The apoptosis rate was measured by AnnexinV-FITC/PI staining using flow cytometry following transfection with indicated plasmids or mock-transfection for 48h. Error bars are SDs for a triplicate analysis of three independent experiments (G&J), or SDs for 15 cells (C&H). All significance analyses were assessed using One-way ANOVA with Dunnett's multiple comparisons test.

promotes their stability and enhances lysosomal function, while desialylation increases their susceptibility to lysosomal degradation, leading to a shorter half-life [49,50]. The HN protein of NDV possesses sialidase activity, making it unique among all viral proteins. This activity allows the HN protein to cleave sialic acid, preventing virus self-aggregation during progeny virus budding [51]. Considering the critical role of sialylation modification in maintaining the stability of LAMPs, we therefore sought to investigate whether HN protein mediates the deglycosylation and degradation of LAMP1 and LAMP2 through its sialidase activity.

First, we evaluated the impact of two sialidase activity inhibitors, Peramivir and Zanamivir, on the HN protein-induced deglycosylation and degradation of LAMP1 and LAMP2. Our results demonstrated that treatment with either Peramivir or Zanamivir significantly inhibited the deglycosylation and degradation of LAMP1 and LAMP2 following the transfection of the HN protein (Fig 8A), as well as reduced LMP levels (Fig 8B and 8C). To further investigate the role of sialidase activity in HN protein-induced LMP, we identified key amino acid sites in the

HN protein of the Herts/33 strain that contribute to its sialidase activity. Based on a previous study [52], we conducted site-directed mutations on ten conservative amino acids located in the catalytic pocket of the HN protein. The results showed that E401D and R481A mutations most significantly reduced the sialidase activity of the HN protein (Fig 8G). Importantly, these mutations effectively prevented the deglycosylation and degradation of LAMP1 and LAMP2 (Fig 8F). The interaction between these mutated HN proteins and LAMP1/LAMP2 was also determined by a co-immunoprecipitation assay. The results revealed that neither the E401D nor R481A mutation affected the interaction between the HN protein and LAMP1 or LAMP2 protein (Fig 8D and 8E). Finally, we investigated the effects of these two mutations on HN protein-induced cellular LMP and apoptosis. Our results revealed that the HN protein mutants exhibited a significantly reduced ability to induce LMP (Fig 8H and 8I) and apoptosis (Fig 8J) compared to the wild-type HN protein.

To further assess the universality of HN protein-induced LMP across diverse NDV strains, we generated plasmids expressing the HN protein using cDNA from six distinct NDV strains. These included lentogenic strains La Sota (GenBank Accession Number: AF077761) and LX (KF494201), mesogenic strains Mukteswar (EF201805.1) and JS/07/04/Pi (FJ766530), and virulent strains F48E8 (FJ436302.1) and Kuwait256 (MK978147). Sialidase activity assays unveiled significantly higher activity in the HN protein derived from virulent and mesogenic strains compared to that from lentogenic strains (S7A Fig). Additionally, Western blotting and confocal microscopy experiments demonstrated that transfection of HeLa cells with expression plasmids from different strains induced deglycosylation and degradation of LAMP1 and LAMP2 (S7B Fig), along with LMP (S7C Fig). Notably, the levels of induction were more pronounced for HN proteins derived from virulent and mesogenic strains (S7B and S7C Fig). These results suggest that the induction of LMP, mediated by the HN protein, demonstrates a degree of universality across distinct NDV strains and is positively correlated with the sialidase activity of the HN protein.

Overall, these findings demonstrated that the sialidase activity of the HN protein is crucial for the deglycosylation and degradation of LAMP1 and LAMP2, ultimately leading to cellular LMP and apoptosis during the infection of diverse NDV strains.

## Discussion

In this study, we aimed to investigate the induction of LMP by NDV infection in cells. Our findings provide compelling evidence that NDV infection triggers extensive LMP in various tumor and avian cells. This is supported by several key observations, including reduced lysosomal acidification, leakage of acridine orange from lysosomes, and translocation of galectin3 to lysosomal membranes. Importantly, we observed that NDV infection promotes the translocation of CTSB and CTSD from the lysosomal lumen to the cytoplasm. The translocation of cathepsins, a consequence of LMP, often leads to different forms of cell death, with apoptosis being the most extensively studied [53]. In the cytosol, cathepsins regulate apoptosis by activating apoptotic proteases and degrading antiapoptotic proteins [54–57]. Previous studies have demonstrated the involvement of cathepsins in cell apoptosis induced by various viruses. For instance, CTSB and cathepsin S have been shown to contribute to apoptosis via caspase activation in Noroviruses [58] and DENV [18] infections, while highly pathogenic human CoVs, including SARS-CoV, MERS-CoV, and SARS-CoV-2, utilize cathepsin L to activate apoptosis and facilitate viral dissemination [59,60]. In our study, we provided evidence that specific inhibitors of CTSB and CTSD, as well as corresponding siRNAs, significantly suppress cell apoptosis and inhibit the cleavage of PARP1, caspase 3, caspase 7, and caspase 9 following NDV infection. These findings strongly support that LMP functions as the key factor for

triggering apoptosis during NDV infection, and this promotional role is, at least partially, via CTSB and CTSD. In addition to regulating cell apoptosis, cathepsins also play multiple roles in viral infection, including promoting viral attachment and entry into target cells, antigen processing and presentation, and facilitating viral progeny release [61]. In our study, we demonstrated that inhibition of either CTSB or CTSD activity suppresses NDV replication, while promotion of the release of these cathepsins by LLoMe enhances NDV replication. Similar effects of cathepsin activity on the positive regulation of viral replication have also been reported in Influenza A virus [62], Hendra virus [63], Nipah virus [64], and Calicivirus [65]. Nevertheless, further investigation is required to fully understand the mechanism by which LMP-induced translocated cathepsins promote NDV replication.

The permeability of the mitochondrial outer membrane is regulated by the Bcl-2 protein family. Previous studies have reported that LMP can induce MOMP by disrupting the balance between pro-apoptotic and anti-apoptotic members of the Bcl-2 family, leading to MOMP and initiating cell apoptosis [3,15]. For example, the antimalarial drug Quinacrine has been shown to promote LMP, resulting in the release of cathepsin L into the cytosol. This, in turn, promotes Bid cleavage, MOMP, cytochrome-c release, and cell death in both in vitro and in vivo models [66]. In our study, we observed a significant loss of MMP, time-dependent degradation of Bcl-2 protein, time-dependent increase in Bid cleavage, mitochondrial translocation of Bax, and release of mitochondrial Cyt C following NDV infection. These findings are consistent with previous studies [26,67,68] and clearly demonstrate that NDV infection induces mitochondrial dysfunction, ultimately resulting in mitochondria-dependent apoptosis. Importantly, we discovered that inhibiting either CTSB or CTSD activity effectively mitigates NDV-induced mitochondria-dependent apoptosis, providing further support for the role of LMP-induced translocated cathepsins in mitochondrial dysfunction and subsequent activation of cell apoptosis.

Aside from viral infection, LMP can also be induced by a wide range of endogenous molecules. Among these, LMP induced by ROS has been extensively studied [11]. Previous studies have demonstrated that elevated levels of ROS play a crucial role in chemically-induced LMP and cell death, including substances such as bupivacaine [69], imiquimod [70], and gentamicin [71]. Furthermore, infections of Ad5 [17] and DENV [18,19] have been reported to induce LMP through promoting the release of ROS. Therefore, investigating the role of ROS in NDV-induced LMP and cell death is of great interest. Our results demonstrate that NDV infection triggers ROS generation. Given that MOMP is a primary source of intracellular ROS, it is plausible to speculate that the increase of ROS is a consequence of NDV-induced MOMP. Additionally, inhibition of either CTSB or CTSD expression significantly reduced NDV-induced ROS levels, indicating that NDV-induced LMP promotes ROS generation. Furthermore, treatment with the antioxidant NAC significantly inhibited NDV-induced LMP, indicating the promotional role of ROS in LMP induction. Considering that NDV-induced LMP promotes MOMP, these findings collectively reveal that ROS plays a critical role in mediating the crosstalk between LMP and MOMP during NDV infection, amplifying LMP and forming a positive feedback loop that exacerbates MOMP.

LAMP1 and LAMP2 are ubiquitously expressed and account for approximately half of the proteins in the lysosomal membrane [72]. Studies have reported that LMP is often accompanied by the degradation of LAMP1 and LAMP2. For example, oncogenes such as K-ras and erbb2 induce the degradation of LAMP1 and LAMP2 in human colon and breast carcinoma cells, rendering the cells more susceptible to photo-oxidation-induced LMP [73]. Treatment with doxorubicin reduces the expression of LAMP1 and LAMP2 proteins, leading to LMP in human umbilical vein endothelial cells [74]. In our study, we demonstrated that NDV infection causes post-transcriptional degradation of both LAMP1 and LAMP2 proteins in various

tumor and avian cells, while it does not significantly affect the expression of other lysosomal membrane proteins including LAMP3, LIMP II, and LAPTM5. Additionally, we observed a significant deglycosylation of LAMP1 and LAMP2 proteins following NDV infection. Considering the critical role of the highly glycosylated luminal domain in preventing the degradation of LAMPs, it is reasonable to speculate that LAMP1 and LAMP2 proteins undergo deglycosylation before degradation. Furthermore, the digestion assay using the glycosidase PNGase F also supports the involvement of protein deglycosylation in the degradation of LAMP1 and LAMP2 after NDV infection. Additionally, based on the results obtained from treatment with different drug inhibitors, we found that the degradation of LAMP1 and LAMP2 depends on lysosomal acidification and the activity of CTSB. Since lysosomal acidification is a necessary condition for cathepsins to exert their cleavage activity, active CTSB may be the key regulator of LAMP1 and LAMP2 degradation. Collectively, these findings suggest that NDV infection induces deglycosylation of LAMP1 and LAMP2 proteins, followed by their degradation mediated by CTSB within the lysosomal lumen, ultimately leading to LMP.

Concerning the specific degradation of LAMP1 and LAMP2 during NDV infection, with no impact on LAMP3, LIMP II, and LAPTM5, the exact reasons remain elusive. However, considering the structural and functional variations among these proteins, two reasons are hypothesized to contribute to the observed differences in protein stability. Firstly, LAMP1 and LAMP2, characterized by single-pass transmembrane structures with similar lengths and 37% amino acid sequence homology, contrast with LAMP3 (four-pass transmembrane), LIMP II (two-pass transmembrane), and LAPTM5 (five-pass transmembrane) [44,75,76]. The more complex structures of LAMP3, LIMP II, and LAPTM5 may impact their interaction with the HN protein. Secondly, LAMP1 and LAMP2, representing nearly half of lysosomal membrane proteins, while LAMP3, LIMP II, and LAPTM5 show comparatively lower expression on lysosomal membranes [76,77]. Additionally, apart from LAPTM5, the glycosylation pattern differs: LAMP1 and LAMP2 possess 24 glycosylation sites, while LAMP3 has 7, and LIMP II has 10 [76]. The abundant expression and multiple glycosylation sites of LAMP1 and LAMP2 likely offer ample "sites" for the HN protein to mediate deglycosylation. Therefore, the unique structural, wide distribution, and glycosylation features of LAMP1 and LAMP2, compared to LAMP3, LIMP II, and LAPTM5, may contribute to the NDV-induced degradation. However, further investigations are essential to unveil the precise underlying mechanisms.

The HN protein of NDV is an envelope glycoprotein that possesses both hemagglutinin and sialidase activities [51]. NDV infection initiates with the binding of the HN protein to the cell surface receptor, primarily sialic acid, through its hemagglutinin activity. In the late stage of viral infection, the sialidase activity of the HN protein prevents virion aggregation and aids viral spread within infected cells. Additionally, studies have indicated that the involvement of the HN protein in apoptosis induction by NDV [45–47]. However, the precise molecular mechanism underlying the pro-apoptotic activity of the HN protein remains poorly identified. In our study, we made a significant discovery that among all of the viral proteins, the HN protein alone can induce the deglycosylation and degradation of LAMP1 and LAMP2, leading to LMP and apoptosis in HeLa cells. Furthermore, our findings provide evidence of a direct interaction between the HN protein and LAMP1 or LAMP2, which triggers the deglycosylation of LAMP1 and LAMP2 through the sialidase activity of the HN protein. These results significantly expand our current knowledge regarding the molecular mechanisms underlying HN protein-mediated apoptosis. Additionally, our results indicate that HN protein-mediated LMP exhibits a certain degree of universality among different NDV strains. However, compared to lentogenic strains, the HN protein of mesogenic and virulent strains appears to induce stronger LMP, due to their higher sialidase activity. This result suggests the potential regulation of NDV virulence by the sialidase activity of the HN protein through the induction of LMP.

Interestingly, a previous study indicates that the neuraminidase (NA) of the H5N1 influenza A virus can induce similar phenotypes, including the induction of deglycosylation and degradation of LAMP1 and LAMP2 proteins, as well as the formation of lysosomal damage and cell death [78]. This suggests that sialidase activity might serve as a crucial "weapon" for virus-induced cell death through the lysosomal pathway. Meanwhile, in our study, we observed that LMP can be induced by the HN proteins from different virulence NDV strains, implying a certain universality in HN protein-mediated LMP. However, in influenza A viruses, the conservation of LMP mediated by NA seems to vary, as the NA of the seasonal H1N1 virus does not exhibit these characteristics [78]. This suggests that, in influenza A viruses, factors other than NA may regulate the formation of LMP, supported by the observation that the NA protein of the H1N1 virus, despite possessing NA activity, cannot induce LMP formation [78]. Therefore, further investigations are needed to determine whether there are other factors involved in regulating LMP induced by NDV infection.

Currently, the specific cellular location and mechanism of interaction between the HN protein and LAMP1/LAMP2 remain unknown. The HN protein of NDV is synthesized in the cytoplasm and undergoes post-translational modifications in the endoplasmic reticulum (ER). Subsequently, the immature HN protein is transported from the ER to the Golgi apparatus. After further modifications, the mature HN protein is then transported from the trans-Golgi network to the cell surface via vesicular transport. The delivery mechanism of LAMP1 and LAMP2 remains poorly understood, with suggestions of indirect plasma membrane mediation or direct intracellular pathway involvement [33]. LAMP1 and LAMP2 are synthesized in the ER, transported through the Golgi complex to the trans-Golgi network, and then follow either the secretory pathway to the plasma membrane before transportation to lysosomes through endocytosis or undergo endocytosis to reach lysosomes directly. Based on our results, it is unlikely that the interaction between LAMP1 or LAMP2 and the NDV HN protein occurs within lysosomes, as no HN protein was detected in the lysosomal fraction during NDV infection (Fig 2A). Considering the involvement of mature proteins and their convergence during transport, it is reasonable to assume that LAMP1 and LAMP2 may interact with the HN protein in the trans-Golgi network or in vesicles during their transportation to the plasma membrane, or in the plasma membrane. Further experimental investigations are required to validate these hypotheses and gain a more comprehensive understanding of the interaction between HN and LAMP1 or LAMP2.

In conclusion, our study demonstrates for the first time that NDV utilizes sialidase activity of the HN protein to hydrolyze the sialic acid residues of LAMP1 and LAMP2, leading to their deglycosylation and subsequent degradation by CTSB within the lysosomal lumen, triggering LMP. Subsequently, CTSB and CTSD translocate to the cytoplasm, ultimately resulting in mitochondria-dependent apoptosis (Fig 9). Our findings unravel the intricate interplay between NDV infection, LMP, and apoptosis, highlighting the potential of targeting these processes for antiviral strategies and the development of NDV-based oncolytic agents.

## Materials and methods

### Cell culture and virus

HeLa (CCL-2), A549 (CCL-185), and DF-1 (CRL-3586) cells were purchased from the ATCC. HD11 cells were kindly provided by Prof. Jiao Xinan (Yangzhou University, Yangzhou, China). The cells were cultured in Dulbecco's modified Eagle's medium (DMEM, Thermo Fisher Scientific, Waltham, USA) supplemented with 10% fetal bovine serum (Thermo Fisher Scientific) and 1% Penicillin-Streptomycin (Thermo Fisher Scientific), at 37˚C, in a 5% $CO_2$ incubator. NDV strain Herts/33 was obtained from Dr. D. J. Alexander (Animal Health and

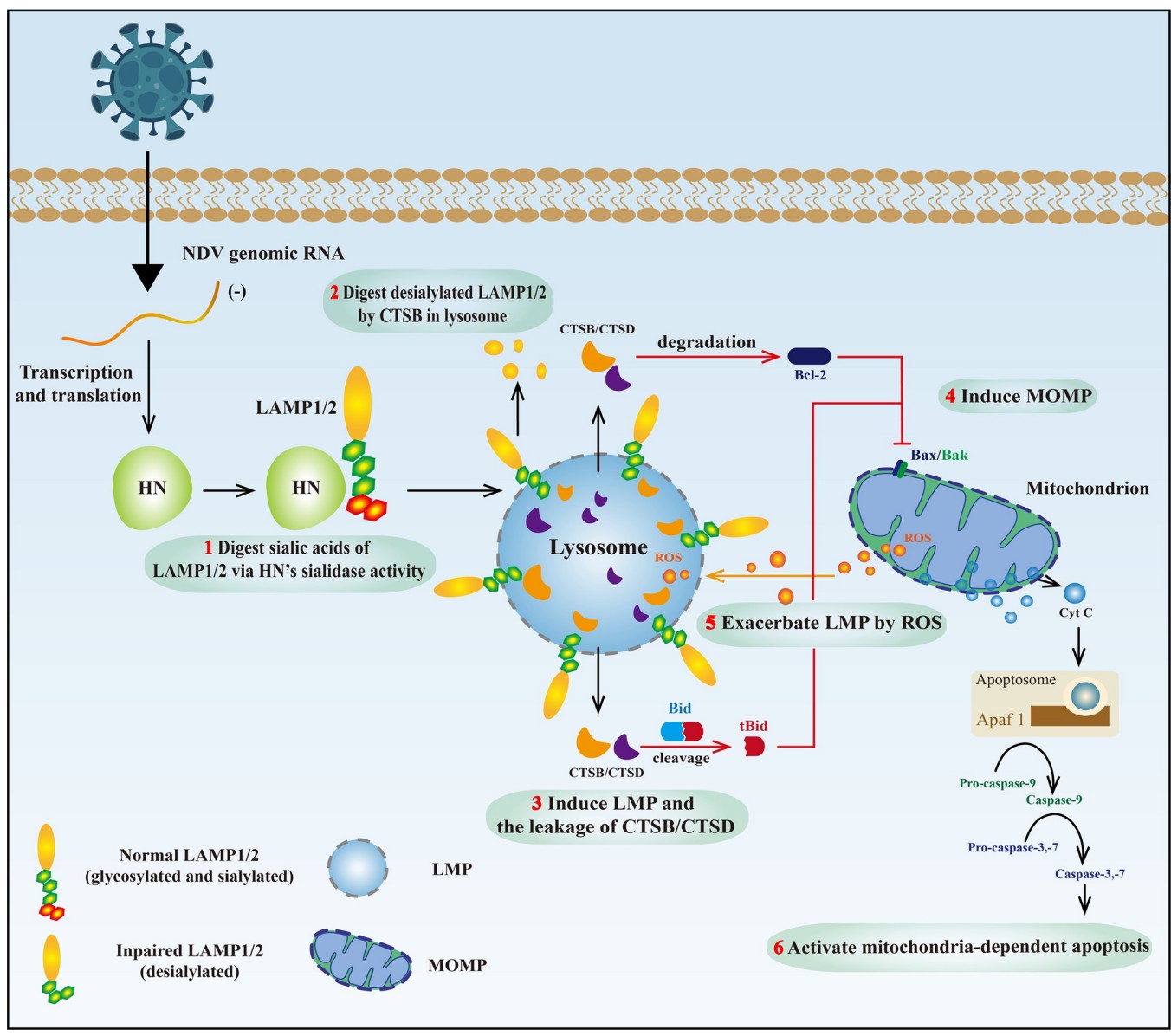

**Fig 9. A mechanism model showing the HN protein of NDV triggers LMP and promotes apoptosis.** NDV enters host cells and utilizes its genome for transcription and translation of the HN protein. The HN protein digests the sialic acid at the end of the glycan chains of LAMP1 and LAMP2 via its sialidase activity, leading to desialylation. The desialylated LAMP1 and LAMP2 undergo deglycosylation and degradation in the lysosome by CTSB, leading to LMP and the leakage of CTSB and CTSD. Consequently, these translocated CTSB and CTSD promote the degradation of Bcl-2, the cleavage of Bid into tBid, and the mitochondrial translocation of Bax, thereby inducing MOMP. This process further exacerbates LMP by inducing ROS release while simultaneously activating mitochondria-dependent apoptosis through the release of Cyt C.

Veterinary Laboratories Agency, UK). The virus was propagated in 9-day-old specific-pathogen-free (SPF) embryonated chicken eggs (Vital River, Beijing, China) via allantoic cavity inoculation.

## Antibodies and reagents

Anti-galectin 3 rabbit monoclonal antibody(sc-32790), anti-LAMP2 mouse monoclonal antibody (sc-18822), anti-LIMP II mouse monoclonal antibody (sc-55571), anti-CTSB mouse

monoclonal antibody (sc-365558), anti-CTSD mouse monoclonal antibody (sc-13148), anti-NDV HN mouse monoclonal antibody (sc-53562) and mouse IgG (sc-2025) were purchased from Santa Cruz Biotechnology, Texas, USA.

Goat anti-rabbit IgG (H+L) HRP-conjugated secondary antibody (HS101-01), goat anti-mouse IgG (H+L) HRP-conjugated secondary antibody (HS201-01), anti-His mouse monoclonal antibody (HT501-01), anti-Flag mouse monoclonal antibody (HT201-01), anti-HA mouse monoclonal antibody (HT301-01), anti-GAPDH mouse monoclonal antibody (HC301-02), anti-β-actin mouse monoclonal antibody (HC201-02), and anti-α-tubulin mouse monoclonal antibody (HC101-02) were purchased from TransGen, Beijing, China.

Anti-LAMP1 mouse monoclonal antibody (9091S), anti-Cleaved Caspase 9 rabbit monoclonal antibody (9505T), anti-Cleaved Caspase-3 rabbit (9661S), anti-Bid rabbit monoclonal antibody (2002T), anti-Cyt C rabbit monoclonal antibody (4280T), and anti-COX IV rabbit monoclonal antibody (4850T) were purchased from Cell Signaling Technology, Danvers, USA.

Goat anti-rabbit IgG H&L (Alexa Fluor 647) secondary antibody (ab150083), goat anti-mouse IgG H&L (Alexa Fluor 488) secondary antibody (ab150117), anti-LAMP2 rabbit monoclonal antibody (ab13524) and anti-Cleaved Caspase 7 rabbit monoclonal antibody (ab256469) were purchased from Abcam, Waltham, USA. Anti-LAMP3 rabbit polyclonal antibody (12632-1-AP), anti-PARP1 mouse monoclonal antibody (66520-1-Ig), anti-Bcl-2 mouse monoclonal antibody (68103-1-Ig) and anti-Bax mouse monoclonal antibody (60267-1-Ig) were purchased from Proteintech, Rosemont, USA.

Anti-LAPTM5 rabbit polyclonal antibody (orb184851) was purchased from Biorbyt, Cambridge, UK. Anti-NP mouse monoclonal antibody was kindly provided from Prof. Ding Chan (Shanghai Veterinary Research Institute, Chinese Academy of Agricultural Sciences, Shanghai, China).

Baf A1 (HY-100558), LLoMe (HY-129905), CA-074 (HY-103350), Pep A (HY-P0018), 3-MA (HY-19312), MG-132 (HY-13259), Zanamivir (HY-13210), Peramivir (HY-17015), CHX (HY-12320) and NH4CL (HY-Y1269) were purchased from MedChemExpress, Shanghai, China. NAC (S0077), PNGase F (20411ES01) was purchased from Yeasen Biotechnology, Shanghai, China.

## Plasmid construction and transfection

All plasmids were named as PCAGGS-gene name-tag (abbreviated as "pCA- gene name-tag"). The primers used in this study are listed in S1 Table. Plasmids were verified by sequencing.

To construct plasmids expressing Flag-tagged viral NP, P, M, F, HN, L, V, and W proteins, the coding sequence (CDS) of each gene was amplified using specific primer pairs and the cDNA of the Herts/33 strain (GenBank Accession Number: AY741404) as a template. The amplified genes were then cloned into the EcoRI/XhoI site of the PCAGGS-Flag-N vector (HG-VPH1041, HonorGene, Changsha, China) using the ClonExpress Ultra One Step Cloning Kit V2 (C116-01, Vazyme, Nanjing, China) according to the manufacturer's instructions. To construct plasmids expressing HN protein of different NDV strains, the cDNA of each strain was served as a template, and specific primers for their respective HN genes were employed to amplify the CDS of the HN gene. The amplified genes were then cloned into the EcoRI/XhoI site of the PCAGGS-Flag-N vector as described above.

For the construction of eight HN protein mutation plasmids, the PCA-HN-Flag was used as a template, and the Mut Express II Fast Mutagenesis Kit V2 (C214-01, Vazyme, Nanjing, China) was used.

To construct plasmids expressing lysosome membrane proteins, the CDS of Homo sapiens LAMP1 (huLAMP1, GenBank Accession Number: NM_005561.4), Homo sapiens LAMP2

(huLAMP2, GenBank Accession Number: NM_002294.3), Gallus gallus LAMP1 (chLAMP1, GenBank Accession Number: NM_205283.3), Gallus gallus LAMP2 (chLAMP2, GenBank Accession Number: NM_001397756.1), Gallus gallus LAMP3 (chLAMP3, GenBank Accession Number: NM_001146132), Gallus gallus LIMP II (chLIMP II, GenBank Accession Number: XM_040670845.2), Gallus gallus LAPTM5 (chLAPTM5, GenBank Accession Number: XM_040690064.2) were PCR amplified using cDNA extracted from HeLa or DF-1 cells as templates. The amplified genes were then cloned into the EcoRI/NotI site of the PCAGGS vector (HG-VPA0057, HonorGene, Changsha, China) using the ClonExpress Ultra One Step Cloning Kit. The chLAMP1, chLAMP2, chLAMP3, chLIMP II, and chLAPTM5 were fused with different tags.

For transfection, cells were transfected at 40% confluency with the corresponding plasmids by the TransIntro EL Transfection Reagent (FT201-01, TransGen) following the manufacturer's instructions.

## RNA interference (RNAi)

siRNAs (Genepharma, Suzhou, China) were utilized to downregulate the expression levels of CTSB, CTSD, LAMP1, and LAMP2 in HeLa cell. A sequence devoid of homology to all genes in human cells was chosen as the siRNA negative control (si-NC). The sequences of siRNAs are shown in S2 Table. HeLa cells were transfected with siRNAs using TransIntro EL Transfection Reagent at a final concentration of 50 nM. The knockdown efficiency was assessed through Western blotting assay as described below.

## NDV infection and chemicals treatment

For NDV infection, HeLa, A549, DF-1, and HD11 cells were absorbed with the Herts/33 strain at the indicated multiplicity of infection (MOI) for 1 h in serum-free DMEM. Then, the unattached viruses were removed, and the cells were washed three times with phosphate-buffered saline (PBS) and cultured in complete medium at 37°C under a 5% $CO_2$ atmosphere. After viral infection, the cell samples were used for further analysis.

For apoptosis and virus replication detection, HeLa or DF-1 cells were treated with indicated chemicals, including LLoMe (500 nM), Ca-074 (20 μM), and Pep A (20 μM) for 3h prior to NDV infection. Viral infection was conducted as described above. The infected cells were cultured in the completed medium containing corresponding chemicals for indicated time points.

To investigate the effect of ROS on NDV-induced LMP, HeLa cells were infected with Herts/33 at 0.01 MOI. Then, the cells were cultured in complete medium containing NAC (2 mM) for 24 h.

To assess the stability of LAMP1 and LAMP2 protein, HeLa cells were infected with Herts/33 at 0.01 MOI for 12h or left uninfected. Then, the cells were treated with CHX (20 μg/mL) for indicated time points.

For detection of the mechanism by which LAMP1 and LAMP2 were degraded, HeLa cells were infected with Herts/33 at 0.01 MOI for 6h. Then, the cells were cultured in the completed medium containing both CHX (20 μg/ml) and NH4CL (50 mM), or Ca-074 (10 μM), or Pep A (10 μM), or 3-MA (5mM), or MG132 (5 μM) or left untreated.

For detection of the role of HN protein sialidase activity in regulating LAMP1 and LAMP2, HeLa cells were transfected with plasmid expressing HN protein. After transfection, the cells were cultured in complete medium containing peramivir (30 μg/ml) or zanamivir (20 μg/ml) for indicated time points.

## Virus growth kinetics

Indicated chemicals-pretreated HeLa or DF-1 cells were infected with Herts/33 strain at 0.01 MOI and constantly treated with corresponding chemicals. The culture supernatants were collected and replaced with an equal volume of fresh media at indicated time points post infection. The viral titers in the supernatants were quantified by $TCID_{50}$ assay in CEF cells as described in a previous study [79].

## Lysosome and mitochondrial extraction

To detect the distribution of CTSB and CTSD in cells, the NDV-infected HeLa cells were washed twice with PBS and collected using a cell scraper. The lysosome and cytoplasmic fractions were separated using a Lysosome Enrichment Kit (BB-31452, Bestbio, Shanghai, China) as described in a previous study [80]. To detect the distribution of the Bax protein, the mitochondria and cytoplasmic fractions were freshly isolated from NDV-infected and/or siRNA-transfected HeLa cells following the protocol of the Mitochondria Isolation Kit (C3601, Beyotime). In brief, HeLa cells were resuspended in a mitochondria extraction reagent provided in the kit. The resuspended cells were then homogenized using a microhomogenizer and subsequently cooled in an ice bath for 15 minutes. After the cooling step, the homogenates underwent centrifugation at 600 g for 10 minutes at 4˚C. The resulting supernatants were carefully collected and subjected to a second round of centrifugation at 11,000 g for 10 minutes at 4˚C. This second centrifugation step allowed for the separation of the cytoplasmic fraction in the supernatant and the mitochondrial fraction in the precipitates. Lysosome, mitochondria, and cytoplasmic fractions were quantified by the BCA method, and 20 μg protein were subjected to Western blotting analysis.

## Western blotting and co-immunoprecipitation

For Western blotting assay, cells were transfected and/or treated as indicated and then washed three times with cold PBS before being lysed by an RIPA buffer (P0013B, Beyotime). The lysates were obtained by centrifugation at 13,000 g for 15 minutes at 4˚C. The supernatants were collected, and protein concentrations were determined using a BCA protein assay kit (P0010, Beyotime). Equal amounts of proteins (20 μg) were separated on 6% or 10% SDS-PAGE gels and transferred to PVDF membranes (FFP28, Beyotime). The PVDF membranes were blocked with 5% skimmed milk for 1 h at room temperature, followed by incubation with the appropriate primary antibodies. After overnight incubation and three washes with TBST, the PVDF membranes were incubated with HRP-conjugated secondary antibodies at room temperature for 1 h. Membranes were then exposed using a BeyoECL Moon reagent (P0018FS, Beyotime) and scanned using a multi-chemiluminescence image analysis system (Tanon, Nanjing, China). The gray values were analyzed using ImageJ software (NIH, Bethesda, MD).

Co-immunoprecipitation was performed using IP lysis buffer (P0013F, Beyotime). Cells were lysed, and the lysates were centrifuged to obtain supernatants. The supernatants were divided into three tubes. One tube, representing the input, was denatured at 95˚C for 10 minutes. The other two tubes, containing equal amounts of supernatants, were used for immunoprecipitation. One tube was incubated with the indicated antibody, while the other tube was incubated with mouse IgG as a control. The antibody or IgG-supernatant mixture was rotated at 4˚C overnight, followed by incubation with 10 μL of Protein A+G Magnetic Beads (P2179M, Beyotime) at 4˚C for 1 h. The beads-antibody protein complex was collected using a magnetic rack and washed five times with ice-cold IP lysis buffer under gentle shaking. The

immunoprecipitated proteins were eluted from the beads by boiling at 95˚C for 10 minutes. Finally, the proteins were subjected to Western blotting analysis as described above.

## Immunofluorescence

Cells seeded on glass coverslips (WHB-24-CS, WHB, Shanghai, China) in 24-well plates were infected with Herts/33 strain at 0.01 MOI for the indicated time. After three washes with TBST, the cells were fixed and permeabilized with ice-cold methanol at −20˚C for 10 min. Subsequently, the cells were blocked with 5% BSA for 1 h at room temperature and incubated overnight at 4˚C with the appropriate primary antibodies. Following that, the cells were incubated with suitable secondary antibodies for 1 h at room temperature and stained with DAPI for 5 minutes to label the cell nuclei. Fluorescent images were captured using a confocal laser scanning microscope (TCS SP8 STED; Leica Microsystems, Wetzlar, Germany) equipped with a 63×/1.40 oil-immersion objective lens. The colocalization coefficient was measured using the 'Manders' Colocalization Coefficients' plugin of ImageJ software.

## Lysotracker red staining

Cells seeded on glass coverslips in 24-well plates were infected and/or treated as indicated. Then, cells were washed 3–5 times with PBS and incubated in complete medium containing 100 nM LysoTracker Red DND-99 (L7528, Thermo Fisher Scientific) at 37˚C. After 1 hour, cells were washed 3–5 times with PBS and fixed with ice-cold methanol at -20˚C for 10 minutes, followed by additional PBS washes. The fixed cells were imaged using a Leica TCS SP8 Inverted Confocal Microscope (Ex/Em = 577/590 nm). For flow cytometry analysis, cells were stained with LysoTracker Red DND-99, trypsinized with 0.05% Trypsin-EDTA, resuspended in 1× PBS, and immediately analyzed using a flow cytometer (CyAn ADP7, BECMAN, Bria, USA). The mean fluorescence intensity was quantified using FlowJo software (Treestar, Ashland, OR).

## Acridine orange staining

Cells seeded on glass coverslips in 24-well plates were infected with Herts/33 strain at 0.01 MOI for the indicated time. Then, cells were washed 3–5 times with PBS and incubated in complete medium containing 1 μg/ml AO dye (318337, Sigma-Aldrich, Beijing, China) for 30 minutes at 37˚C. After 3–5 times washing with PBS, the AO red (Ex/Em = 555/617 nm) and AO green (Ex/Em = 490/528 nm) fluorescence were evaluated immediately with TCS SP8 STED confocal laser scanning microscope.

## Measurement of MMP

Cells cultured in 24-well plates were infected and/or transfected as indicated. After different treatment, the MMP of treated cells was measured by an MMP Detection Kit (C2006, Beyotime) according to the manufacturer's instructions. For fluorescence microscope observation, cells were rinsed with pre-warmed PBS for one time and incubated with JC-1 staining solution (5 μg/mL) for 20 min at 37˚C. Cells were then rinsed twice with cold JC-1 staining buffer, and fluorescence intensity of both JC-1 monomers (Ex/Em = 514/529 nm) and JC-1 aggregates (Ex/Em = 585/590 nm) were detected using an inverted fluorescent microscope (AXIOVERT A1, Zeiss, Oberkochen, Germany). For flow cytometry analysis, treated cells were harvested and rinsed with PBS twice. After corresponding treatment, the fluorescence intensity of JC-1 monomers and JC-1 aggregates were analyzed using CyAn ADP7 flow cytometer.

## CTSB and CTSD activity assay

Cells were infected with Herts/33 at 0.01 MOI for indicated time points or mock-infected. The lysosome and cytoplasmic fractions were extracted as described above. Cytoplasmic CTSB and CTSD activities were measured using a reaction buffer provided in the CTSB (ab65300, Abcam) and CTSD (ab65302, Abcam) Activity Assay Kit according to the manufacture's protocol, respectively. Measure output on a microplate reader (Synergy 2, BioTek, Vermont, USA) using a fluorometric assay (Ex/Em = 400/505 nm for CTSB, Ex/Em = 328/460 nm for CTSD). The obtained fluorescence intensity was normalized to the concentration of individual protein.

## Determination of intracellular ROS levels

Cells were infected and/or transfected as indicated. After different treatment, the intracellular ROS levels were determined by ROS Assay Kit (S0033S, Beyotime) following the manufacturer's instructions. Briefly, cells were collected and incubated in the diluted DCFH-DA (10 μM) solution. During incubation of the cells at 37˚C, gently invert the mixture every 3–5 minutes to ensure proper contact between the probe and the cells. Twenty minutes later, wash the cells three times with serum-free cell culture medium to remove any residual DCFH-DA that did not enter the cells. Then, the cells were applied for flow cytometry analysis immediately (Ex/ Em = 488/525 nm).

## Apoptosis assay

The apoptosis ratio was measured by AnnexinV-FITC/PI Cell Apoptosis Detection Kit (FA101-02, TransGen) according to the manufacturer's instructions. Briefly, cells were trypsinized by non-EDTA trypsin and collected by centrifugation at 500 g, 4˚C for 5 minutes. Then, cells were washed thrice with PBS and resuspended in 100 μL pre-chilled 1×Annexin V Binding Buffer, supplemented with 5 μL Annexin V-FITC and 5 μL PI. Cells were incubated at room temperature for 15 minutes in the dark. After incubation, 400 μL Annexin-binding buffer was added, and samples were immediately analyzed in CyAn ADP7 flow cytometer.

## qRT-PCR analysis

For detection of the mRNA levels of lysosomal membrane proteins, cells were infected with Herts/33 strain as indicated. The mRNA was extracted from the cells using the EasyPure RNA kit (ER101-01, TransGen). Then, cDNA synthesis and qRT-PCR reaction were performed using TransScript Green One-Step qRT-PCR Super Mix (AQ211-01, TransGen) according to the manufacturer's instructions on LightCycler 480 (Roche, Basel, Switzerland). The specific primers used for qPCR were listed in S3 Table. GAPDH was used as a normalized control, and the mRNA levels were calculated by the $2^{-\Delta\Delta Ct}$ method.

## Statistical analysis

All data were presented as means ± SD as indicated. Student's t-test, one-way and two-way ANOVA tests were used for the analysis of studies where appropriate. All statistical analyses and calculations were carried out using GraphPad Prism software (V8.0, San Diego, CA, USA). A P value of less than 0.05 was regarded as statistically significant. NS means no significant difference, $^*P < 0.05$, $^{**}P < 0.01$, $^{***}P < 0.001$, $^{****}P < 0.0001$.

## Supporting information

**S1 Fig. NDV infection induces LMP in different cell lines.** (A-C) A549 (A), DF-1 (B), and HD11 (C) cells were infected with Herts/33 at 0.01 MOI for the indicated time points. Torin 1 (1 μM, 4h) and Baf A1 (1 μM, 4h) were used as positive and negative controls, respectively. Lysotracker fluorescence signals were detected by flow cytometry after staining the cells with Lysotracker (100 nM) for 1h. (D) A549, DF-1, and HD11 cells were infected with Herts/33 at 0.01 MOI for 24h or mock-infected. Then, AO green and red fluorescence signals were detected by confocal microscopy following incubation with AO (1 μg/mL) for 30 min. Scale bars, 20 μm. (E) A549 cells were infected with Herts/33 at 0.01 MOI for the indicated time points or mock-infected. The cells were then stained with rabbit anti-galectin 3 (green) and mouse anti-LAMP1 (red) antibodies and were observed by confocal microscopy. Scale bars, 20 μm.
(TIF)

**S2 Fig. NDV infection induces the leakage of CTSB and CTSD from the lysosomal lumen to the cytoplasm in A549 cells.** (A&C) A549 cells were infected with Herts/33 at 0.01 MOI for the indicated time points or mock-infected. The colocalization of CTSB (A) or CTSD (C) with LAMP3 was detected using confocal microscopy. Scale bars, 20 μm. (B&D) Manders' Colocalization Coefficients of CTSB (B) or CTSD (D) with LAMP3 were quantified by ImageJ software. Error bars represent SDs for 15 cells (B&D). Significance was assessed using one-way ANOVA with Dunnett's multiple comparison test.
(TIF)

**S3 Fig. LMP facilitates NDV replication in HeLa and DF-1 cells.** (A&B) HeLa (A) and DF-1 (B) cells were pretreated with LLoMe (500 nM), Ca-074 (5 μM), Pep A (5 μM), or mock-treated for 3h, respectively. Subsequently, the cells were infected with Herts/33 at 0.01 MOI along with the corresponding chemicals in the cell culture medium. Culture supernatants were collected at the indicated time points, and viral titers were determined by the 50% tissue culture infectious dose (TCID50) assay. (C&D) HeLa (C) and DF-1 (D) cells were pretreated and infected as described above. Protein levels of NP and HN were assessed by Western blotting after 24h of NDV infection. The relative intensity ratio of the indicated proteins, normalized to GAPDH, was quantified using ImageJ software and is presented on the left side of the panel. All error bars are SDs for a triplicate analysis of three independent experiments. Significance was assessed using Two-way ANOVA with Dunnett's multiple comparisons test (A&B) or one-way ANOVA with Dunnett's multiple comparison test (C&D).
(TIF)

**S4 Fig. Verification of the antioxidant effect of NAC.** (A) HeLa cells were treated with NAC (2 mM) for 24h after absorption with Herts/33 at 0.01 MOI for 1h. Intracellular ROS levels were measured by flow cytometry using DCFH-DA staining. (B) The quantitative data are presented as means ± SD (n = 3). Significance was assessed using one-way ANOVA with Dunnett's multiple comparison test.
(TIF)

**S5 Fig. The effect of NDV infection on the expression of indicated lysosome associated proteins in A549, DF-1, and HD11 cell lines.** (A) A549 cells were infected with Herts/33 at 0.01 MOI or mock-infected. After the indicated time points, the protein levels of indicated lysosome membrane proteins were detected by Western blotting assay. (B&C) DF-1 cells were co-transfected with plasmids expressing avian LAMP1 (His tag) and LAMP2 (Flag tag) (B or co-transfected with plasmids expressing avian LAMP3 (His tag), LIMP II (Flag tag) and

LAPTM5 (HA tag) (C). Forty-eight hours after transfection, the cells were infected with Herts/ 33 at 0.01 MOI or mock-infected for 24h. Protein levels of indicated proteins were then detected by Western blotting assay. (D-F) HeLa (D), A549 (E) and DF-1 (F) cells were infected with Herts/33 at 0.01 MOI or mock-infected. After the indicated time points, the mRNA levels of indicated lysosome membrane proteins were detected by qRT-PCR assay. The mRNA levels were normalized to GAPDH and calculated using the $2-\Delta\Delta Ct$ method. All the quantitative data are presented as means ± SD (n = 3). Significance was assessed using Two-way ANOVA with Dunnett's multiple comparisons test.
(TIF)

**S6 Fig. HN protein induces apoptosis in HeLa cells.** HeLa cells were transfected with a plasmid expressing HN protein (1.6 μg), or empty vector plasmid (1.6 μg), or mock-transfected. The apoptosis rate was measured by AnnexinV-FITC/PI staining using flow cytometry after transfection for 48h. Quantitation of apoptosis rate is calculated as means ± SD (n = 3) and shown in the right panel. Significance was assessed using One-way ANOVA with Dunnett's multiple comparisons test.
(TIF)

**S7 Fig. The induction of LMP mediated by the HN protein exhibits a degree of universality across distinct NDV strains.** (A&B&C) HeLa cells were transfected with indicated plasmids expressing HN protein of distinct NDV strains (1.6 μg) or mock-transfected for 48h. (A) The sialidase activity was measured using the Neuraminidase Assay Kit. (B) Protein levels of indicated proteins were detected by Western blotting assay. GAPDH was used as a normalized control. (C) The LMP levels were observed by confocal microscopy using anti-galectin 3 (green) and anti-LAMP1 (red) antibodies. Scale bars, 20 μm. Manders' Colocalization Coefficients of galectin 3 with LAMP1 were quantified by ImageJ software and shown on the right. Error bars represent SDs for triplicate analyses of three independent experiments (A), or SDs for 15 cells (C). All significance analyses were assessed using One-way ANOVA with Dunnett's multiple comparisons test.
(TIF)

**S1 Table. Primers for plasmids construction.**
(XLSX)

**S2 Table. Sequence for siRNA.**
(XLSX)

**S3 Table. Primers for qRT-PCR.**
(XLSX)

## Acknowledgments

We want to thank all of our colleagues for their input and many valuable suggestions.

## Author Contributions

**Conceptualization:** Yu Chen, Chan Ding, Shunlin Hu, Xiufan Liu.

**Funding acquisition:** Yu Chen.

**Investigation:** Yu Chen, Chunxuan Wang, Jiajun Han, Zhenyu Yang, Xiaolong Lu.

**Methodology:** Yu Chen, Shanshan Zhu, Tianxing Liao, Zenglei Hu, Jiao Hu.

**Writing – original draft:** Yu Chen.

**Writing – review & editing:** Xiaoquan Wang, Min Gu, Ruyi Gao, Kaituo Liu, Xiaowen Liu, Chan Ding, Shunlin Hu, Xiufan Liu.

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
