## [Decision Letter · Decision Letter 0]

21 Aug 2023

Dear Prof. Liu,

Thank you very much for submitting your manuscript "The HN protein of Newcastle disease virus induces cell apoptosis through the induction of lysosomal membrane permeabilization" for consideration at PLOS Pathogens. As with all papers reviewed by the journal, your manuscript was reviewed by members of the editorial board and by several independent reviewers. In light of the reviews (below this email), we would like to invite the resubmission of a significantly-revised version that takes into account the reviewers' comments.

We cannot make any decision about publication until we have seen the revised manuscript and your response to the reviewers' comments. Your revised manuscript is also likely to be sent to reviewers for further evaluation.

Sincerely,

Peter Palese

Academic Editor

PLOS Pathogens

Meike Dittmann

Section Editor

PLOS Pathogens

Kasturi Haldar

Editor-in-Chief

PLOS Pathogens

orcid.org/0000-0001-5065-158X

Michael Malim

Editor-in-Chief

PLOS Pathogens

orcid.org/0000-0002-7699-2064

Reviewer's Responses to Questions

**Part I - Summary**

Reviewer #1: The authors investigated the mechanism of tumor and avian cell apoptosis induced by a virulent NDV infection. The authors showed that NDV infection can reduce lysosome acidification by increasing LMP, resulting of CTSB and CTSD being released into the cytosol, generating ROS and induce mitochondria dysfunction, which leads to mitochondria-dependent apoptosis. The authors also showed the expression of only the HN protein of NDV deglycosylated LAMP1 and LAMP2 via its sialidase activity and therefore destabilized both proteins (reduced protein expression), which could lead to increased LMP. By IP and IF, HN appeared to bind and co-localize with LAMP1 and LAMP2. This is an interesting study adding value to understand the pathogenesis of NDV. However, the following comments should be sufficiently addressed.

Reviewer #2: In the manuscript “The HN protein of Newcastle disease virus induces cell apoptosis through the induction of lysosomal membrane permeabilization” Yu Chen and colleagues elucidate how the Newcastle disease virus induces apoptosis via lysosomal damage. The authors describe how the viral Hemagglutinin neuraminidase (HN) protein digests sialic acids at the end of glycan chains of lysosome-associated membrane protein 1 (LAMP1) and LAMP2, making them vulnerable to degradation and triggering lysosomal membrane permeabilization (LMP). Furthermore, they show how the apoptotic cascades activated by leakage of lysosomal proteins cathepsin B and D end up inducing mitochondria-dependent apoptosis.

This study is nicely constructed, elegantly telling a cohesive story, with extensive amounts of data and very thorough. However, some necessary controls, references and explanations are missing. These concerns need to be addressed to make this a high quality publication.

**Part II – Major Issues: Key Experiments Required for Acceptance**

Reviewer #1: -What happens in cells that is less susceptible to NDV infection? Such as cells that can only support a single round of virus replication? Did the authors observed increased LMP?

-Does lentogenic NDV also has the same effects? It was previously reported that the HN protein of NDV determines tropism of virulency (PMID: 15047833). The HN protein of the Lentogenic strains such as la sota seems to have less NA activity as compared to virulent strain BC. The authors should compare avirulent strain and virulent strain by either infection or transfection to see if this is universal to all NDV HN proteins.

-Figure 3A and 3E, do CTSB and CTSD inhibitors or CTSB and CTSD knockdown reduce NDV infectivity? Please show the expression levels of viral protein(s), such as NP.

-Figure 3E and 3Q, the authors should show protein expression in mock infected cells with knockdown of CTSB and CTSB as a control.

-The phenotype in figure 1-4 was very clear when infecting cells with NDV at MOI of 0.01 for 24 hours. However, the reduction of LAMP1 expression and deglycosylation in figure 5C was only significant when infecting cells at MOI of 1 for 24 hours. On the other hand, LAMP2 seems to be more affected by NDV infection at MOI of 0.01 than LAMP1. But in Figure 5F, after 12 hour of infection at MOI of 0.01, LAMP1 is already deglycosylated and degraded? And F5I is 6 hours after infection at MOI of 0.01? Please clarify.

Reviewer #2: 1. Ju and colleagues showed 2015 in JVI something very similar for the Neuraminidase of Influenza A virus. This paper needs to be referenced.

2. Figure 3E: a non-target siRNA control is needed (similar to how it is done in Figure 6K)

3. Figure 5I and the corresponding result section need to address some inconsistencies:

a. Figure 5F and I: In Figure 5F, NDV 12hpi and CHX treatment, LAMP1 and LAMP2 are fully degraded. In Figure 5I, LAMP1 and LAMP2 have strong bands 12h post CHX treatment and NDV infection (mock other treatments). Why the difference in expression between the experiments?

b. Figure 5I: All the bands of samples treated with CHX appear to be lower than samples not treated with CHX (h0). Does CHX lead to deglycosylation? Please add possible explanation

c. Figure 5I: PepA, 3-MA and MG132 appear to increase degradation of LAMP1 and LAMP2 compared to mock treated cells. Please address

**Part III – Minor Issues: Editorial and Data Presentation Modifications**

Reviewer #1: -The role of torin 1 in figure 1A should be mentioned in the result

-To mutate 10 conserved amino acids in HN neuraminidase active sites, why not do alanine scanning for all 10 residues? For example, I175L doesn’t seem to be a big change.

Reviewer #2: 1. Some bar graphs show data points, some do not. Preferably, show data points for all bar graphs.

2. Define abbreviations: line 169 si-CTSB

Line 372 Ad5 and DENV

MOI

3. The authors should briefly discuss why LAMP3 stays stable upon NDV infection whereas LAMP1 and LAMP2 are being degraded (as depicted in Figure 2A). What makes LAMP3 different?

4. Figure 2C and 2F: It appears that the amount of cytoplasmic cathepsin B decreases 36hpi compared to 24hpi while the activity of cytoplasmic cathepsin B increased in the same time span. Possible explanation?

5. Figure 3 is very busy. Maybe split figure into two or move some of the data to supplementary figures.

6. Figure 3T: Labelling Mitochondrial fraction appears doubled

Figure 3O, P: The NS are shifted

7. Figure 5E: LAPTM5 is decreasing in the 24hpi column compared to the 1 MOI column?

8. Figure 6A and B: In Figure 6A HN leads to deglycosylation, in Figure 6B HN leads to deglycosylation AND degradation. Where does the difference come from? Transfected with different amounts of plasmid?

9. Figure 6E and F are too small

10. The labelling of western blots in supplementary figure 5 is inconsistent with the other figures, using anti-LAMP1/LAMP2/Flag and so forth instead of just LAMP1 etc.

11. Pepstatin A is an aspartic acid protease inhibitor, not a cathepsin D specific inhibitor (line 160).

PLOS authors have the option to publish the peer review history of their article (what does this mean?). If published, this will include your full peer review and any attached files.

Reviewer #1: No

Reviewer #2: No
---

## [Decision Letter · Decision Letter 1]

27 Dec 2023

Dear Prof. Liu,

Thank you very much for submitting your manuscript "The HN protein of Newcastle disease virus induces cell apoptosis through the induction of lysosomal membrane permeabilization" for consideration at PLOS Pathogens. As with all papers reviewed by the journal, your manuscript was reviewed by members of the editorial board and by several independent reviewers. The reviewers appreciated the attention to an important topic. Based on the reviews, we are likely to accept this manuscript for publication, providing that you modify the manuscript according to the review recommendations.

Please, address minor minor comment of reviewer #1

Sincerely,

Peter Palese

Academic Editor

PLOS Pathogens

Meike Dittmann

Section Editor

PLOS Pathogens

Kasturi Haldar

Editor-in-Chief

PLOS Pathogens

orcid.org/0000-0001-5065-158X

Michael Malim

Editor-in-Chief

PLOS Pathogens

orcid.org/0000-0002-7699-2064

Please, address minor minor comment of reviewer #1

Reviewer Comments (if any, and for reference):

Reviewer's Responses to Questions

**Part I - Summary**

Reviewer #1: The authors addressed most of my comments sufficiently

Reviewer #2: The authors have addressed my concerns and revised the manuscript accordingly.

**Part II – Major Issues: Key Experiments Required for Acceptance**

Reviewer #1: no major issues

Reviewer #2: (No Response)

**Part III – Minor Issues: Editorial and Data Presentation Modifications**

Reviewer #1: please clarify what is si-NC in the method section since this control was newly added

Reviewer #2: (No Response)

PLOS authors have the option to publish the peer review history of their article (what does this mean?). If published, this will include your full peer review and any attached files.

Reviewer #1: No

Reviewer #2: No

Figure Files:

Data Requirements:

Reproducibility:

References:

---

## [Editor Report · Decision Letter 2]

2 Jan 2024

Dear Prof. Liu,

Thank you very much for submitting your manuscript "The HN protein of Newcastle disease virus induces cell apoptosis through the induction of lysosomal membrane permeabilization" for consideration at PLOS Pathogens. As with all papers reviewed by the journal, your manuscript was reviewed by members of the editorial board and by several independent reviewers. The reviewers appreciated the attention to an important topic. Based on the reviews, we are likely to accept this manuscript for publication, providing that you modify the manuscript according to the review recommendations.

Please, address minor point of reviewer #1

Sincerely,

Peter Palese

Academic Editor

PLOS Pathogens

Meike Dittmann

Section Editor

PLOS Pathogens

Kasturi Haldar

Editor-in-Chief

PLOS Pathogens

orcid.org/0000-0001-5065-158X

Michael Malim

Editor-in-Chief

PLOS Pathogens

orcid.org/0000-0002-7699-2064

Please, address minor point of reviewer #1

Reviewer Comments (if any, and for reference):

Figure Files:

Data Requirements:

Reproducibility:

References:

---

## [Editor Report · Decision Letter 3]

17 Jan 2024

Dear Prof. Liu,

We are pleased to inform you that your manuscript 'The HN protein of Newcastle disease virus induces cell apoptosis through the induction of lysosomal membrane permeabilization' has been provisionally accepted for publication in PLOS Pathogens.

Best regards,

Peter Palese

Academic Editor

PLOS Pathogens

Meike Dittmann

Section Editor

PLOS Pathogens

Kasturi Haldar

Editor-in-Chief

PLOS Pathogens

orcid.org/0000-0001-5065-158X

Michael Malim

Editor-in-Chief

PLOS Pathogens

orcid.org/0000-0002-7699-2064
---

## [Editor Report · Acceptance letter]

30 Jan 2024

Dear Prof. Liu,

We are delighted to inform you that your manuscript, "The HN protein of Newcastle disease virus induces cell apoptosis through the induction of lysosomal membrane permeabilization," has been formally accepted for publication in PLOS Pathogens.

Best regards,

Michael Malim

Editor-in-Chief

PLOS Pathogens

orcid.org/0000-0002-7699-2064